# Multiple mechanisms link prestimulus neural oscillations to sensory responses

**Luca Iemi[1,2,3]\*, Niko A Busch[4,5], Annamaria Laudini[6], Saskia Haegens[1,7], Jason Samaha[8], Arno Villringer[2,6], Vadim V Nikulin[2,3,9,10]\***

[1]Department of Neurological Surgery, Columbia University College of Physicians and Surgeons, New York City, United States; [2]Department of Neurology, Max Planck Institute for Human Cognitive and Brain Sciences, Leipzig, Germany; [3]Centre for Cognition and Decision Making, Institute for Cognitive Neuroscience, National Research University Higher School of Economics, Moscow, Russian Federation; [4]Institute of Psychology, University of Münster, Münster, Germany; [5]Otto Creutzfeldt Center for Cognitive and Behavioral Neuroscience, University of Münster, Münster, Germany; [6]Berlin School of Mind and Brain, Humboldt-Universität zu Berlin, Berlin, Germany; [7]Donders Institute for Brain, Cognition and Behaviour, Radboud University Nijmegen, Nijmegen, Netherlands; [8]Department of Psychology, University of California, Santa Cruz, Santa Cruz, United States; [9]Department of Neurology, Charité-Universitätsmedizin Berlin, Berlin, Germany; [10]Bernstein Center for Computational Neuroscience, Berlin, Germany

**\*For correspondence:**
luca.iemi@gmail.com (LI);
nikulin@cbs.mpg.de (VVN)

**Abstract** Spontaneous fluctuations of neural activity may explain why sensory responses vary across repeated presentations of the same physical stimulus. To test this hypothesis, we recorded electroencephalography in humans during stimulation with identical visual stimuli and analyzed how prestimulus neural oscillations modulate different stages of sensory processing reflected by distinct components of the event-related potential (ERP). We found that strong prestimulus alpha- and beta-band power resulted in a suppression of early ERP components (C1 and N150) and in an amplification of late components (after 0.4 s), even after controlling for fluctuations in 1/f aperiodic signal and sleepiness. Whereas functional inhibition of sensory processing underlies the reduction of early ERP responses, we found that the modulation of non-zero-mean oscillations (baseline shift) accounted for the amplification of late responses. Distinguishing between these two mechanisms is crucial for understanding how internal brain states modulate the processing of incoming sensory information.
DOI: https://doi.org/10.7554/eLife.43620.001

## Introduction

The brain generates complex patterns of neural activity even in the absence of sensory input or tasks. This activity is referred to as 'spontaneous', 'endogenous', or 'prestimulus', as opposed to activity triggered by and, thus, following sensory events. Numerous studies have shown that such spontaneous neural activity can explain a substantial amount of the trial-by-trial variability in perceptual and cognitive performance (e.g., *Haegens et al., 2011*; *Myers et al., 2014*; *Iemi et al., 2017*) and that abnormalities in spontaneous neural activity serve as biomarkers for neuropathologies (e.g., in schizophrenia, Parkison's disease, and Autism Spectrum Disorder; *Uhlhaas and Singer, 2010*; *McCarthy et al., 2011*; *Simon and Wallace, 2016*) and aging (*Voytek et al., 2015*). Yet, the mechanisms by which spontaneous neural activity impacts the processing of sensory information remain unknown.

**eLife digest** Give a computer the same input and you should get back the same response every time. But give a human brain the same sensory input and you will see a range of different responses. This is because the brain's response to sensory input depends not only on the properties of the input, but also on its own internal state at the time when the input is processed. Even in the absence of any input, the brain generates complex patterns of spontaneous activity. Fluctuations in this activity affect how the brain responds to the outside world.

The electrical activity in the brain – both spontaneous and in response to sensory input – can be measured using electrodes close to the scalp: this measurement is referred to as electroencephalography, or EEG. Spontaneous brain activity takes the form of rhythmic waves, also known as oscillations. In a person who is awake and relaxed, the EEG consists mainly of slow oscillations called alpha and beta waves. Sensory input, such as an image or a sound, triggers changes in brain activity that can be seen in the EEG. This EEG response is called an event-related potential, or ERP, and consists of a characteristic pattern of peaks and troughs in the EEG.

To find out how spontaneous brain activity affects ERPs, Iemi et al. presented images of black and white checkerboards to healthy volunteers. The results showed that the ERP looked different if the stimulus occurred during strong alpha and beta waves. The early part of the ERP – which occurs between 80 and 200 milliseconds after the onset of the stimulus – decreased in size, presumably because it was inhibited by strong alpha and beta waves. In contrast, the later part of the ERP – which occurs more than 400 milliseconds after stimulus onset – increased in size. This paradox is accounted for by a newly recognized feature of the oscillations, namely that they fluctuate around a non-zero value of the EEG. Thus, two different mechanisms contributed to these opposite changes.

The findings add to our understanding of how spontaneous brain activity influences how we perceive the world around us. Furthermore, spontaneous brain activity differs in a number of disorders, including schizophrenia and autism. Understanding how spontaneous neural oscillations affect how the brain processes information from the senses could provide new insights into these conditions.

DOI: https://doi.org/10.7554/eLife.43620.002

This study aims to clarify how spontaneous fluctuations of prestimulus brain states, reflected by the power of low-frequency oscillations in the $\alpha$- and $\beta$-bands (8–30 Hz), affect the trial-by-trial variability in the amplitude of sensory event-related potentials (ERPs). The mechanisms underlying the effect of prestimulus power on ERP amplitudes are currently unknown, partly because previous studies have been inconsistent regarding the latency and even the directionality of this effect. Specifically, several studies found that prestimulus $\alpha$-band power suppresses the amplitude of early ERP components (<0.200 s: *Rahn and Başar, 1993*; *Roberts et al., 2014*; *Becker et al., 2008*; *Başar and Stampfer, 1985*; *Jasiukaitis and Hakerem, 1988*), whereas other studies found that prestimulus $\alpha$-band power enhances the amplitude of late ERP components (>0.200 s: *Dockree et al., 2007*; *Becker et al., 2008*; *Roberts et al., 2014*; *Başar and Stampfer, 1985*; *Jasiukaitis and Hakerem, 1988*; *Barry et al., 2000*). In this study, we addressed this issue by considering how prestimulus power affects the mechanisms of ERP generation at different latencies: namely, additive and baseline-shift mechanisms.

ERP components occurring during the early time window (<0.200 s) are thought to be generated by an activation of sensory brain areas adding on top of ongoing activity (additive mechanism: *Bijma et al., 2003*; *Shah et al., 2004*; *Mäkinen et al., 2005*; *Mazaheri and Jensen, 2006*). Invasive studies in non-human primates demonstrated that early ERP components are associated with an increase in the magnitude of multi-unit activity (MUA) in sensory areas (*Kraut et al., 1985*; *Schroeder et al., 1990*; *Schroeder et al., 1991*; *Schroeder et al., 1998*; *Shah et al., 2004*; *Lakatos et al., 2007*), presumably as a result of membrane depolarization due to excitatory synaptic activation (*Schroeder et al., 1995*). Non-invasive studies in humans showed that early ERP components (e.g., C1) are associated with an increase in the hemodynamic fMRI signal in visual areas (*Di Russo et al., 2002*), which may reflect an additive sensory response. Low-frequency neural oscillations are thought to set the state of the neural system for information processing (*Klimesch et al.,*

*2007*; *Jensen and Mazaheri, 2010*; *Mathewson et al., 2011*; *Spitzer and Haegens, 2017*), which in turn may modulate the generation of additive ERP components. In particular, numerous studies have demonstrated that states of strong ongoing $\alpha$- and $\beta$-band oscillations reflect a state of functional inhibition, indexed by a reduction of neuronal excitability (e.g., single-unit activity: *Haegens et al., 2011*; *Watson et al., 2018*; MUA: *van Kerkoerle et al., 2014*; *Becker et al., 2015*; ongoing $\gamma$-band power: *Spaak et al., 2012*; hemodynamic fMRI signal: *Goldman et al., 2002*; *Becker et al., 2011*; *Scheeringa et al., 2011*; *Harvey et al., 2013*; *Mayhew et al., 2013*) and of subjective perception (e.g., conservative perceptual bias: *Limbach and Corballis, 2016*; *Iemi et al., 2017*; *Craddock et al., 2017*; *Iemi and Busch, 2018*; lower perceptual confidence: *Samaha et al., 2017b*; lower visibility ratings: *Benwell et al., 2017*). Accordingly, prestimulus low-frequency oscillations in the $\alpha$- and $\beta$-bands may exert an inhibitory effect on the additive mechanism of ERP generation: that is, states of strong prestimulus power may suppress the activation of sensory areas, resulting in an attenuation of the amplitude of additive ERP components during the early time window (*Figure 1*).

While early ERP components are likely to be generated primarily through an additive mechanism, late ERP components can have contributions from both additive and baseline-shift mechanisms (where 'baseline' denotes the mean offset of the signal, rather than prestimulus activity). According to the baseline-shift account, the slow ERP component, which becomes visible during the late time window (>0.200 s), is generated by a modulation of ongoing oscillatory power, rather than by an additive response (*Nikulin et al., 2007*; *Nikulin et al., 2010a*; *Mazaheri and Jensen, 2008*; *Mazaheri and Jensen, 2010*; *van Dijk et al., 2010*). The effect of the baseline-shift mechanism on the relationship between prestimulus power and ERP amplitude has never been tested. In fact, it is generally assumed (and not even questioned) that neural oscillations are symmetrical around the zero line of the signal. Accordingly, trial averaging is expected to eliminate non-phase-locked oscillations due to phase cancellation (assuming a random phase distribution over trials), thereby resulting in a signal baseline with a zero mean. It follows that a modulation of zero-mean oscillations by stimuli/tasks leaves the signal baseline unaffected (*Figure 1A*).

In contrast to this traditional view, recent studies (*Nikulin et al., 2007*; *Nikulin et al., 2010a*; *Mazaheri and Jensen, 2008*; *van Dijk et al., 2010*; *Schalk, 2015*; *Cole and Voytek, 2017*) proposed that neural oscillations do not vary symmetrically around the zero line of the signal, but rather around a non-zero offset/mean (*Figure 1B*). Accordingly, trial averaging does not eliminate non-phase-locked oscillations with a non-zero mean. As a consequence, any amplitude modulation of oscillations with a non-zero mean is expected to change the signal baseline (baseline shift), and will therefore affect the ERP amplitude. Specifically, during the event-related desynchronization (ERD) of low-frequency oscillations, the power suppression is expected to cause a slow shift of the signal baseline toward the zero line. Subtracting the prestimulus non-zero baseline from the post-stimulus signal creates a slow shift, which mirrors the spatio-temporal profile of the ERD. In particular, an ERD of oscillations with a negative non-zero-mean is expected to generate an upward slow shift of the ERP (and viceversa). The idea that the ERD contributes to the generation of the slow ERP component, implies that the larger the ERD, the stronger the slow ERP component. Accordingly, we predicted that states of strong prestimulus power would yield a strong ERD (*Min et al., 2007*; *Becker et al., 2008*; *Tenke et al., 2015*; *Benwell et al., 2018*), resulting in an enhancement of the slow ERP component during the late time window.

To summarize, states of strong prestimulus power are expected: (1) to suppress the amplitude of additive ERP components during the early time window (via functional inhibition); and (2) to amplify the late ERP component, generated by an event-related modulation of non-zero-mean oscillations (via baseline shift). To test these predictions, we recorded electroencephalography (EEG) in human participants during rest and during stimulation with identical high-contrast checkerboard stimuli and analyzed the relationship between ERPs, ongoing and event-related oscillations. We find that the effects of prestimulus power on early and late ERP components are consistent with the functional inhibition and baseline-shift accounts, respectively. Taken together, these results largely resolve apparent inconsistencies in previous literature and specify two distinct mechanisms by which prestimulus neural oscillations modulate visual ERP components.

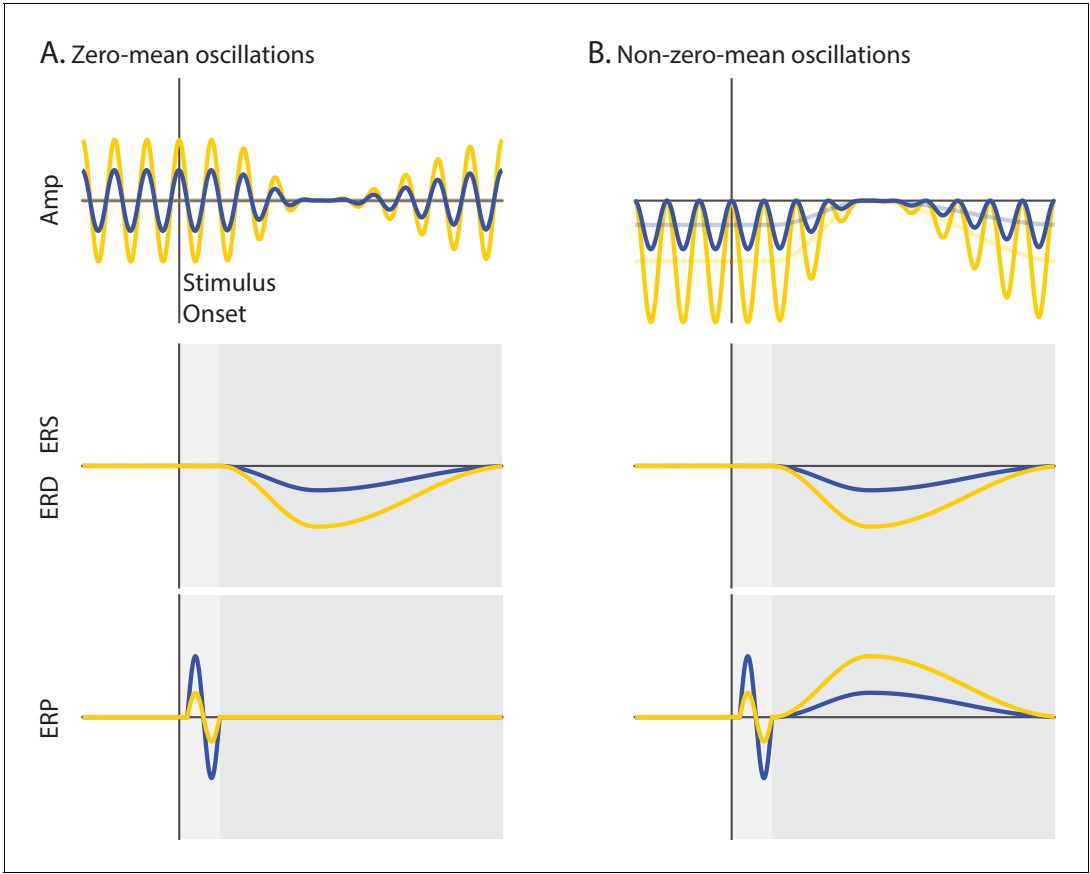

**Figure 1.** Schematic overview of the mechanisms of functional inhibition and baseline shift. Ongoing activity (Amplitude), event-related oscillations (ERS/ERD) and potentials (ERP) are illustrated in upper, middle and lower panels, respectively. The vertical line indicates stimulus onset, while the horizontal line indicates zero signal strength. Yellow and blue represent states of strong and weak prestimulus power, respectively. (**A**) Non-phase-locked ongoing oscillatory activity with a zero-mean. The oscillations are symmetrical relative to the zero line of the signal (A upper panel). (**B**) Non-phase-locked ongoing oscillatory activity with a non-zero-mean. The oscillations are asymmetrical relative to the zero line of the signal. The signal baseline is characterized by a negative offset (opaque lines). The stronger the power of these oscillations, the stronger the negative offset of the signal baseline (B upper panel). During event-related desynchronization (ERD), the ongoing oscillations are suppressed to the zero line of the signal. This implies that the stronger the prestimulus power, the stronger the ERD (A/B middle panels). Trial averaging of zero-mean oscillations eliminates prestimulus oscillatory activity that is not time-locked to the stimulus because opposite oscillatory phases cancel out. This results in baseline signal at the zero line, which is unaffected by ERD. Therefore, an ERD of zero-mean oscillations does not generate the slow ERP component during the late time window because there is no baseline shift for these oscillations (dark gray; A lower panel). Trial averaging of non-zero-mean oscillations does not eliminate non-phase locked ongoing activity. This results in a prestimulus baseline signal with an offset relative to the zero line. During the ERD, the baseline of the signal gradually approaches the zero line of the signal. When the post-stimulus signal is corrected with the prestimulus non-zero baseline, a slow shift of the ERP signal appears, mirroring the ERD time-course. Specifically, an ERD of negative (positive) non-zero mean oscillations shifts the signal upward (downward), generating the slow ERP component of positive (negative) polarity. Crucially, the stronger the prestimulus power, the stronger the ERD, and as a consequence, the stronger the slow shift of the ERP. The baseline-shift account predicts a positive relationship between prestimulus power and the amplitude of the slow ERP during the late time window (dark gray; B lower panel). According to the functional inhibition account, strong prestimulus power attenuates the amplitude of the additive ERP components. This account predicts a negative relationship between prestimulus power and the amplitude of ERP components during the early time window (light gray; A/B lower panels).

DOI: https://doi.org/10.7554/eLife.43620.003

The following source data is available for figure 1:

**Source data 1.** Simulations.

DOI: https://doi.org/10.7554/eLife.43620.004

## Results

### Event-related potentials

The experiment included stimulation trials with high-contrast checkerboard stimuli presented in the lower (LVF; *Figure 2A*, left panel) or upper (UVF; *Figure 2A*, middle panel) visual field with equal probability, and fixation-only trials without any checkerboard stimulus (Fix; *Figure 2A*, right panel). All trials included a change of the central fixation mark at the time of stimulus presentation (see Materials and methods for details). For each participant we quantified the ERP at the electrode with peak activity between 0.055 and 0.090 s after stimulus onset, reflecting the C1 component which indicates initial afferent activity in primary visual cortex (*Di Russo et al., 2002*).

On stimulation trials, the C1 component peaked on average at 0.079 s (SEM = 0.001) and at 0.078 s (SEM = 0.001) for LVF and UVF stimuli respectively, with a maximum at occipito-parietal electrodes (*Figure 2B*, left and middle panels). The comparison of C1 amplitudes at individual peak electrodes on LVF (M = 10.157 µV ; SEM = 0.918) and UVF (M = −10.567 µV; SEM = 1.058) trials revealed the expected polarity reversal, confirming that this component originates from initial afferent activity in early visual areas (*Di Russo et al., 2002*; *Di Russo et al., 2003*; *Bao et al., 2010*). Following the C1, we observed a N150 component peaking between 0.100 and 0.200 s relative to stimulus onset, with an occipital topography and a consistently negative polarity for both LVF and UVF stimuli. The N150 was followed by a late deflection in the time range between 0.200 and 0.600 s relative to stimulus onset, with a parietal topography and consistent positive polarity for both LVF and UVF stimuli.

As expected, on Fix trials no C1 and N150 components were detected in the ERP at individual C1-peak electrodes for LVF and UVF stimuli (mean amplitude in the C1 time window: −0.044 µV, SEM = 0.196). Fix trials showed a late positive deflection with similar timing and topography as on stimulation trials (*Figure 2B*, right panel).

### Event-related oscillations

For each participant we estimated the event-related synchronization (ERS) and desynchronization (ERD) at frequencies between 5 and 30 Hz and at each electrode and time point of the post-stimulus window (0–0.900 s). For the group-level statistical analysis, we used cluster permutation tests to determine at which time, frequency and electrode the ERS/ERD was significantly different from 0 across participants. On LVF trials, the statistical test revealed a negative cluster (p<0.001) indicating ERD, spanning time points from 0 to 0.900 s relative to stimulus onset, frequencies between 6 and 30 Hz, and all 64 electrodes (*Figure 2C*, left panel). The most negative t-statistic was found at 20 Hz, 0.234 s, and at electrode P7. A positive cluster (p=0.041) indicating ERS spanned time points from 0 to 0.900 s relative to stimulus onset, frequencies between 5 and 8 Hz, and all 64 electrodes. The most positive t-statistic was found at 5 Hz, 0.097 s, and at electrode P7.

On UVF trials, the statistical test revealed a negative cluster (p<0.001) indicating ERD, spanning time points from 0 to 0.900 s relative to stimulus onset, frequencies between 6 and 30 Hz, and all 64 electrodes (*Figure 2C*, middle panel). The most negative t-statistic was found at 20 Hz, 0.234 s, and at electrode P7. This test also found two positive clusters indicating ERS. The first positive cluster (p=0.032) spanned time points from 0 to 0.900 s relative to stimulus onset, frequencies between 5 and 8 Hz, and all 64 electrodes. Within this cluster, the most positive t-statistic was found at 5 Hz, 0.152 s, and at electrode P2. The second positive cluster (p=0.034) spanned time points from 0.49 to 0.900 s relative to stimulus onset, frequencies between 13 and 30 Hz, and 62 electrodes. Within this cluster, the most positive t-statistic was found at 17 Hz, 0.648 s, and at electrode FT7.

For Fix trials, the statistical test revealed one negative cluster (p<0.001) indicating ERD, spanning time points from 0 to 0.900 s relative to fixation-target onset, frequencies between 5 and 30 Hz, and all 64 electrodes (*Figure 2C*, right panel). On Fix trials, the most negative t-statistic was found at 20 Hz, 0.214 s, and at electrode PO8.

### Evidence for functional inhibition

The functional inhibition account predicts that states of strong prestimulus power reflect neural inhibition, resulting in reduced amplitude specifically of early ERP components generated by the additive mechanism. To test for this mechanism, we classified trials in five bins based on single-trial

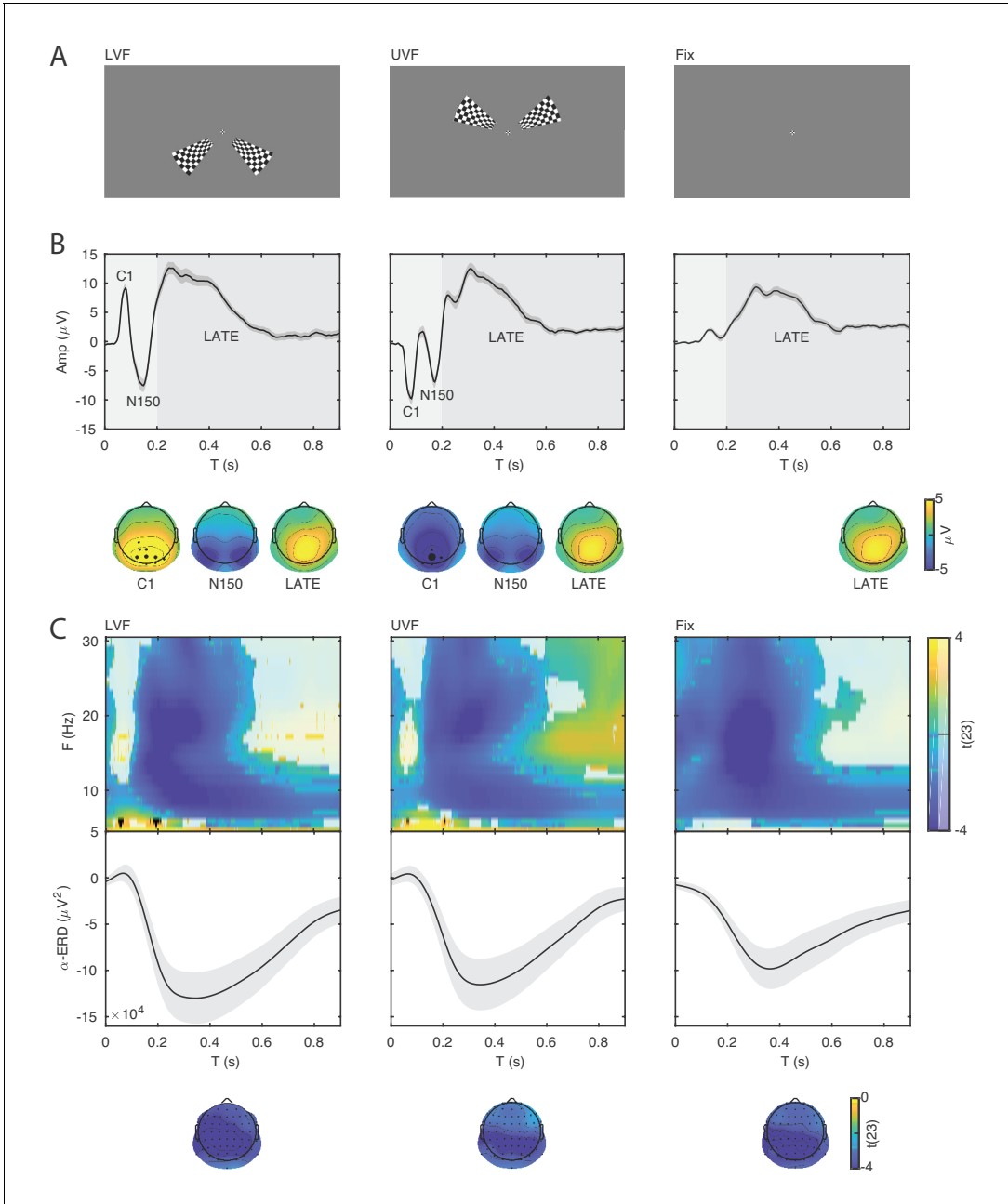

**Figure 2.** Event-related potentials and oscillations. (**A**) The stimuli consisted of bilateral checkerboard wedges specifically designed to elicit the C1 component of the visual ERP. The stimuli appeared in the lower (left panel, LVF), upper visual field (middle panel, UVF), or in no field (right panel, Fix) with equal probability. Across trials, the participants were instructed to discriminate a central target during stimulus presentation. (**B**) Event-related potentials (ERP) were calculated for the subject-specific electrode with C1-peak activity. Additive ERP components are visible during the early time window (< 0.200 s, light grey). The C1 component is the earliest component of the visual ERP and is polarity-reversed across fields of stimulation (LVF vs. UVF). The C1 topography illustrates the ERP amplitude averaged at the subject-specific time point of peak activity between 0.055 and 0.090 s. The size of the electrodes in the topography indicates the frequency of the C1-peak electrode in the sample of participants. The C1 is followed by the N150, peaking between 0.130 and 0.180 s relative to stimulus onset. The N150 topography illustrates the ERP amplitude averaged between 0.130 and 0.180 s. Fix trials do not show any robust additive components during the early time window. The slow component of the ERP is visible during the late time window (> 0.200 s, dark gray). The topography of this late component illustrates the ERP amplitude averaged between 0.200 and 0.900 s. This late ERP component is present in all trial types. Time 0 indicates stimulus onset. (**C**) Group-level t-statistics maps of event-related oscillations. Negative values (blue) indicate significant power suppression across participants (ERD), while positive values (yellow) indicate significant power enhancement across participants (ERS). The maps are averaged across electrodes of the significant clusters, and masked by a final alpha of 0.05 using two-sided cluster permutation testing. The topography illustrates the group-level t-statistics averaged for the α frequency band (8–12 Hz) at the time point of

*Figure 2 continued on next page*

*Figure 2 continued*

most negative t-statistics. The bottom insets illustrate the group-level α-band ERD time course at occipital electrodes. The ERD is present in all trial types. Time 0 s indicates stimulus onset. Source data: the original source data are available at DOI: https://doi.org/10.5061/dryad.nm4241p.

DOI: https://doi.org/10.7554/eLife.43620.005

estimates of oscillatory power at each electrode and each frequency between 5 and 30 Hz averaged over the 0.500 s prestimulus window (i.e., total-band power, *Figure 3C*) and compared the ERP amplitude between the strongest and weakest power bin. These total-band power estimates reflect a mixture of both periodic (i.e., oscillations) and aperiodic signals (i.e., 1/f 'background' noise; *Voytek et al., 2015*, see *Figure 3C/D* and *Appendix 1—figure 1*). Therefore, to determine whether ERP differences between total-band power bins were indeed due to oscillatory activity, we repeated the binning analysis using single-trial estimates of the periodic signal (i.e., aperiodic-adjusted power; see Materials and methods for details).

On LVF trials, a statistical test comparing ERP amplitudes during the early time window (<0.200 s) on trials with strong vs weak prestimulus power found a significant negative cluster (p=0.015). This indicates that the ERP amplitude in a time range containing the C1 (0.043 to 0.121 s) was weaker (i.e., less positive) on trials with strong prestimulus power between 8 and 28 Hz, and at all 64 electrodes (*Figure 3A*, left panel). The most negative t-statistic was found at 10 Hz, 0.078 s, and at electrode P4 (t(23)=−6.030). At this time-frequency-electrode point, this effect was corroborated by the aperiodic-adjusted analysis (t(23)=−3.634; FDR-corrected p=0.003), demonstrating that the ERP amplitude was indeed modulated by oscillatory power (*Appendix 1—figure 1C*). Furthermore, the statistical test revealed a significant positive cluster (p<0.001), indicating that the ERP amplitude in a time range containing the N150 (0.090 to 0.200 s) was weaker (i.e., less negative) on trials with strong prestimulus power between 5 and 24 Hz, and at all 64 electrodes (*Figure 3A*, left panel). The most positive t-statistic was found at 9 Hz, 0.145 ms, and at electrode P1 (total-band power: t(23)=7.940; aperiodic-adjusted power: t(23)=5.381; FDR-corrected p<0.001, *Appendix 1—figure 1C*).

On UVF trials, the statistical test during the early time window revealed two significant positive clusters. The first cluster (p<0.001) indicated that the ERP amplitude in a time range containing the C1 (0.02 to 0.113 s) was weaker (i.e., less negative) on trials with strong prestimulus power between 5 and 22 Hz, and at all 64 electrodes (*Figure 3A*, middle panel). The most positive t-statistic was found at 13 Hz, 0.082 ms, and at electrode PO4 (total-band power: t(23)=8.365; aperiodic-adjusted power: t(23)=2.315; FDR-corrected p=0.045, *Appendix 1—figure 1C*). The second cluster indicated that the ERP amplitude in a time range containing the N150 (0.125 to 0.200 s) was weaker (i.e., less negative) on trials with strong prestimulus power between 5 and 25 Hz, and at all 64 electrodes. The most positive t-statistic was found at 10 Hz, 0.168 ms, and at electrode F2 (total-band power: t(23)=8.544; aperiodic-adjusted power: t(23)=5.785; FDR-corrected p<0.001, *Appendix 1—figure 1C*).

For Fix trials, the statistical test during the early time window found no significant clusters (<0.200 s, *Figure 3A*, right panel).

Taken together, the results on early ERP components show that ERP amplitude on stimulation trials is attenuated during states of strong prestimulus power, regardless of the polarity of the components. This provides evidence for the functional inhibition mechanism underlying the modulatory effect of prestimulus power on the early ERP components.

## Evidence for baseline shift

The baseline-shift account predicts that states of strong prestimulus oscillations with a non-zero mean are followed by strong post-stimulus power suppression (ERD), resulting in a greater ERP amplitude specifically during the late time window. To demonstrate that the late ERP component was generated by a baseline shift, it is necessary to establish that: (1) the ongoing oscillations have a non-zero mean; (2) the non-zero mean and the late ERP component have opposite polarity; and that (3) the ERD magnitude is associated with the amplitude of the late ERP component.

To demonstrate the non-zero mean property of ongoing oscillations, we computed the Baseline Shift Index (*BSI*: *Nikulin et al., 2007*; *Nikulin et al., 2010a*) and the Amplitude Fluctuation Asymmetry Index (*AFAI*: *Mazaheri and Jensen, 2008*). For each participant we estimated *BSI* and *AFAI* from resting-state oscillations for each electrode and frequency between 5 and 30 Hz, and then tested

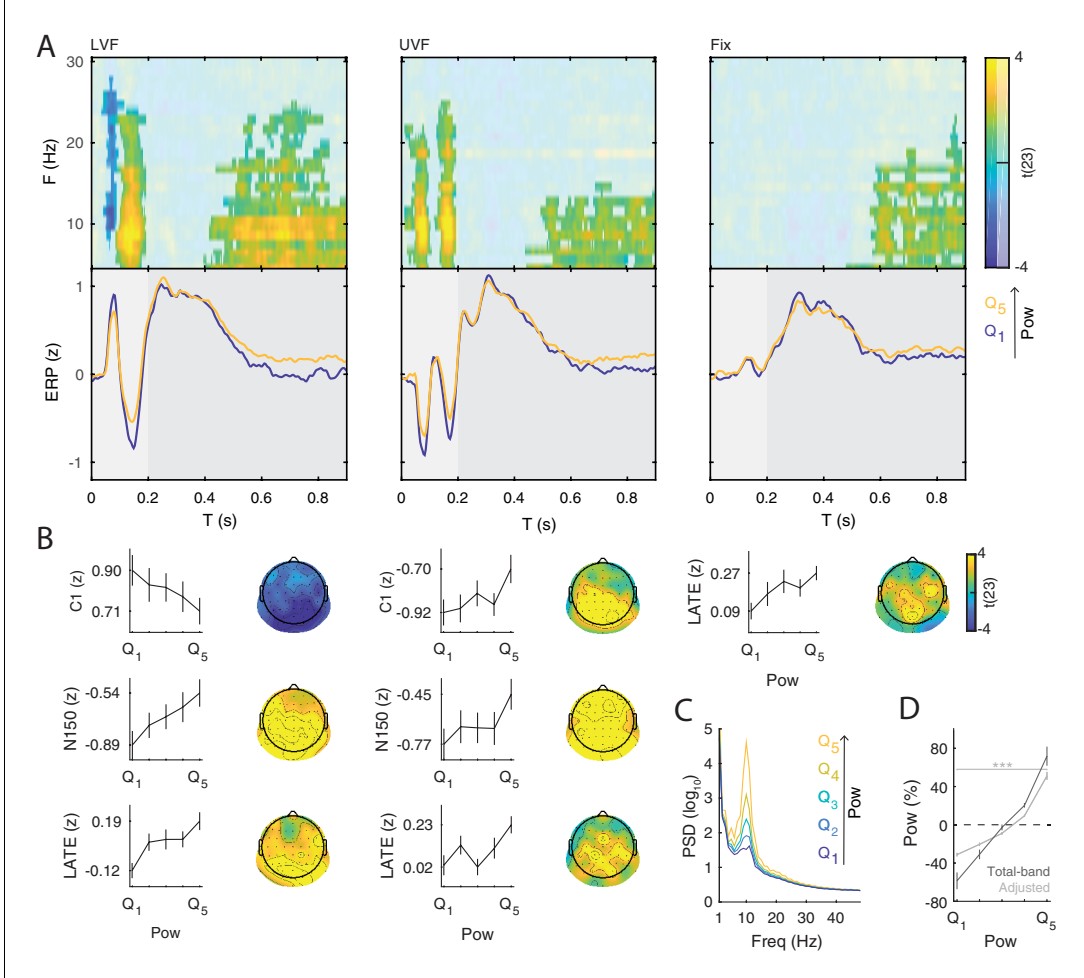

**Figure 3.** Prestimulus power differently modulates the amplitude of early and late event-related potentials (ERP). (A) Group-level t-statistics maps of the difference in ERP amplitude between states of weak ($Q_1$) and strong ($Q_5$) prestimulus power on lower visual field (left panel, LVF), upper visual field (middle panel, UVF), and fixation-only trials (right panel, Fix). Positive (yellow) and negative t-statistics values (blue) indicate that ERP amplitude/voltage becomes more positive or more negative during states of strong prestimulus power, respectively. Accordingly, positive t-statistics values indicate an enhancement of positive ERP components, and a dampening of negative ERP components, while negative t-statistics values indicate an enhancement of negative ERP components, and a dampening of positive ERP components. The maps are averaged across electrodes of the significant cluster, and masked by a final alpha of 0.025 using separate two-sided cluster permutation testing for early and late time windows. Note that the x-axis refers to post-stimulus ERP time, while the y-axis refers to the frequency of prestimulus oscillatory power. Bottom insets: visualization of the normalized ERP time course separately for states of strong ($Q_5$, yellow) and weak ($Q_1$, blue) prestimulus power. This was computed at the electrode and frequency of most positive/negative t-statistics during the C1 time window on stimulation trials and during the late time window on Fix trials. The ERP is characterized by distinct components occurring during the early (<0.200 s: C1 and N150, light gray) and late time windows (>0.200 s, dark gray). Time 0 indicates stimulus/fixation-target onset. (B) Group-average normalized ERP amplitude on trials sorted from weak ($Q_1$) to strong ($Q_5$) prestimulus power, calculated at the ERP time point and the prestimulus-power electrode and frequency of most positive/negative t-statistics. Error bars indicate ± SEM. The topographies show the positive and negative t-statistics at the time point and frequency of most positive/negative t-statistic for early (C1 and N150) and late ERP components. Black dots represent electrodes comprising the significant clusters. States of strong prestimulus power are followed by a reduction of the amplitude of additive ERP components during the early time window (consistent with the functional inhibition account), as well as by an enhancement of the slow ERP component during the late time window (consistent with the baseline-shift account). (C) Group-average prestimulus total-band power spectrum shown separately for the five bins sorted from weak ($Q_1$) to strong ($Q_5$) total-band power at frequencies and electrodes of most positive/negative t-statistics. (D) Group-average percentage change in total-band (dark) and aperiodic-adjusted (light) power sorted from weak ($Q_1$) to strong ($Q_5$) total-band power at frequencies and electrodes of most positive/negative t-statistics. ∗∗∗ indicates FDR-corrected p<0.001 for a one-sample two-sided t-test comparing values between $Q_1$ and $Q_5$.

DOI: https://doi.org/10.7554/eLife.43620.006

whether these indices were significantly different from 0 across participants using cluster permutation tests. For $AFAI$, we found a significant negative cluster (p-value < 0.001) between 5 and 30 Hz with a parietal, occipital and central topography (*Figure 4A/B*), indicating a stronger modulation of the troughs relative to the peaks, resulting in a negative mean. For $BSI$, we found a significant negative cluster (p-value < 0.001) between 5 and 21 Hz with a parietal, occipital and central topography (*Figure 4A/B*), similar to $AFAI$. A negative BSI indicates that strong oscillatory power corresponds to a more negative value of the low-pass filtered signal, as expected in the presence of oscillations with a negative mean. Taken together, the results on $BSI$ and $AFAI$ provide evidence for a non-zero (negative) mean of resting-state low-frequency oscillations. It is important to note that the late ERP component had a positive polarity in all trial types (*Figure 1B*), which is expected as a result of ERD of oscillations with a negative mean (*Nikulin et al., 2007*; *Nikulin et al., 2010a*; *Mazaheri and Jensen, 2008*).

Next, we analyzed the relationship between the ERD magnitude and the ERP amplitude during the late time window (>0.200 s). We compared the amplitude of the late ERP between groups of trials of weak and strong ERD estimated at each frequency and electrode. For the group-level statistical analysis, we used cluster permutation tests to determine significant ERP differences across ERP time points, and ERD electrodes and frequencies. The statistical test on LVF trials revealed one significant positive cluster (p<0.001), indicating that the late ERP (0.200–0.900 s) was greater during states of stronger ERD at frequencies between 5 and 30 Hz, and at all 64 electrodes (*Figure 5A/B*, left panel). The most positive t-statistic was found at 8 Hz, 0.266 s, and at electrode POz. The statistical test on UVF trials revealed two significant positive clusters, indicating that the late ERP (cluster 1: 0.336–0.900 s; cluster 2: 0.200–0.328 s) was greater during states of strong ERD at frequencies between 5 and 30 Hz, and at all 64 electrodes (*Figure 5A/B*, middle panel). The most positive t-statistic was found at 19 Hz, 0.258 s, and at T8 electrode. The statistical test on Fix trials revealed one significant positive cluster (p<0.001), indicating that the late ERP (0.488–0.900 s) was greater during states of strong ERD at frequencies between 5 and 30 Hz, and at all 64 electrodes (*Figure 5A/B*, right panel). The most positive t-statistic value was found at 13 Hz, 0.637 s, and at electrode PO8.

Taken together, these results show that states of stronger ERD were associated with a more positive deflection of the late ERP component, consistent with the baseline-shift account.

After demonstrating that the ERD magnitude correlates with the late ERP amplitude, we determined whether the ERD magnitude was, in turn, related to prestimulus power, as predicted

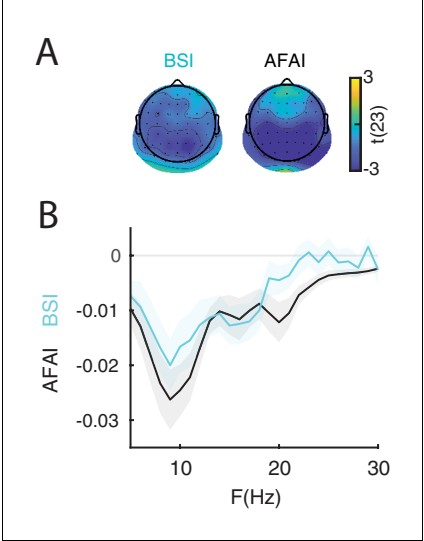

**Figure 4.** The non-zero mean property of resting-state neural oscillations is a prerequisite for the baseline-shift mechanism. The late component of the ERP is generated by a baseline shift occurring during event-related desynchronization (ERD) of non-zero mean oscillations. (**A**) The amplitude fluctuation asymmetry index ($AFAI$) is quantified as the normalized difference between the variance of the peaks and troughs of an oscillatory signal during resting state. $AFAI$<0 (blue) indicates a stronger modulation of the troughs relative to the peaks, consistent with a negative mean. The topography represents the group-level t-statistics of $AFAI$: a significant negative cluster was found at frequencies between 5 and 30 Hz and with an occipito-parietal peak. The baseline shift index ($BSI$) is quantified as the correlation between the oscillatory envelope at a certain frequency and low-pass filtered EEG signal at 3 Hz during resting-state. $BSI$<0 (blue) indicate a negative relationship between oscillatory envelope and low-pass filtered signal, consistent with negative mean. The topography represents the group-level t-statistics of $BSI$: a significant negative cluster was found at frequencies between 5 and 21 Hz and with an occipito-parietal peak. Black dots represent electrodes comprising the significant clusters. (**B**) Comparison between group-level $AFAI$ (black) and $BSI$ (blue), averaged across respective cluster electrodes, and shown for each frequency. Shaded areas indicate ± SEM. These results indicate the presence of a negative oscillatory mean, consistent with the baseline-shift account.

DOI: https://doi.org/10.7554/eLife.43620.007

by the baseline-shift account. To this end, we compared the ERD magnitude (at the subject-specific C1 electrode) between groups of trials of weak and strong prestimulus power estimated for each frequency and electrode, separately for each trial type. We found that strong low-frequency prestimulus oscillations were associated with strong ERD in all trial types (*Figure 6*). Note that this result is expected due to the circularity in estimating ERD and pre-stimulus power. Interestingly, we found that the poststimulus power is similar across different prestimulus $\alpha$-band bins (*Figure 6C*), suggesting that the stimulus suppressed $\alpha$-band oscillations of different magnitudes to approximately the same level.

After demonstrating that the late ERP amplitude correlates with the ERD magnitude, and that the ERD magnitude in turn correlates with prestimulus power, we tested whether prestimulus power was directly correlated with the amplitude of the late ERP component. To this end, we compared the late ERP amplitude between groups of trials with weak and strong prestimulus power estimated for each frequency and electrode. For the group-level statistical analysis, we used cluster permutation tests to determine significant differences across ERP time points, prestimulus-power frequencies, and electrodes.

The statistical test during the late time window revealed a significant, sustained, and positive cluster in each trial type, indicating that the late ERP component was amplified during states of strong prestimulus power.

On LVF trials, the significant positive cluster (p<0.001) spanned time points from 0.402 to 0.900 s, frequencies between 5 and 25 Hz, and all 64 electrodes (*Figure 3A/B*, left panel). The most positive t-statistic was found at 11 Hz, 0.676 s, and at electrode POz (total-band power: t(23)=6.769; aperiodic-adjusted power: t(23)=3.004; FDR-corrected p=0.014, *Appendix 1—figure 1C*). On UVF trials, the significant positive cluster (p=0.004) spanned time points from 0.445 to 0.900 s relative to stimulus onset, frequencies between 5 and 15 Hz, and all 64 electrodes (*Figure 3A/B*, middle panel). The most positive t-statistic was found at 5 Hz, 0.648 s, and at electrode CP1 (total-band power: t(23) =7.600; aperiodic-adjusted power: t(23)=2.528; FDR-corrected p=0.037, *Appendix 1—figure 1C*). On Fix trials, the significant positive cluster (p<0.001) spanned time points from 0.484 to 0.900 s relative to fixation-target onset, frequencies between 5 and 23 Hz, and all 64 electrodes (*Figure 3A/B*, right panel). The most positive t-statistic was found at 7 Hz, 0.781 s, and at electrode POz (total-band power: t(23)=7.528; aperiodic-adjusted power: t(23)=1.881; FDR-corrected p=0.109, *Appendix 1—figure 1C*).

Taken together, these results show that: (1) the late ERP component is generated by a baseline shift during the ERD of non-zero mean oscillations; (2) states of strong prestimulus power are followed by strong ERD, which manifests as an enhancement of the late ERP component.

## Evidence against a confound by sleepiness

To ensure that the relationship between prestimulus power and ERP amplitude was not simply an epiphenomenon of time-varying variables such as sleepiness, we analyzed the scores of a subjective sleepiness questionnaire that participants filled in at the end of every block (Karolinska Sleepiness Scale, KSS: *Kaida et al., 2006*). First, at the single-subject level, we computed the correlation between prestimulus oscillatory power and KSS rating. At the group-level, we tested whether these correlations were significantly different from 0. We found significant positive clusters for frequencies below 18 Hz and with a widespread topography in each trial type (*Appendix 1—figure 2A*). This result indicates that the stronger the prestimulus power, the higher the subjective sleepiness. Within each participant we removed the contribution of sleepiness to the trial-by-trial estimates of oscillatory power and repeated the power-ERP analysis with these corrected power estimates. The results of this re-analysis (*Appendix 1—figure 2B*) were virtually identical to the ones obtained with raw power estimates (*Figure 3A*), suggesting that the effects we observed were not confounded by sleepiness.

## Discussion

Numerous studies have suggested that spontaneous fluctuations of prestimulus brain states can account for the trial-by-trial variability in sensory responses. Specifically, the power of low-frequency neural oscillations in the $\alpha$- and $\beta$-bands (8–30 Hz) was found to be associated with ERP amplitude in visual, auditory, and somatosensory modality (e.g., *Becker et al., 2008*; *Jones et al., 2009*;

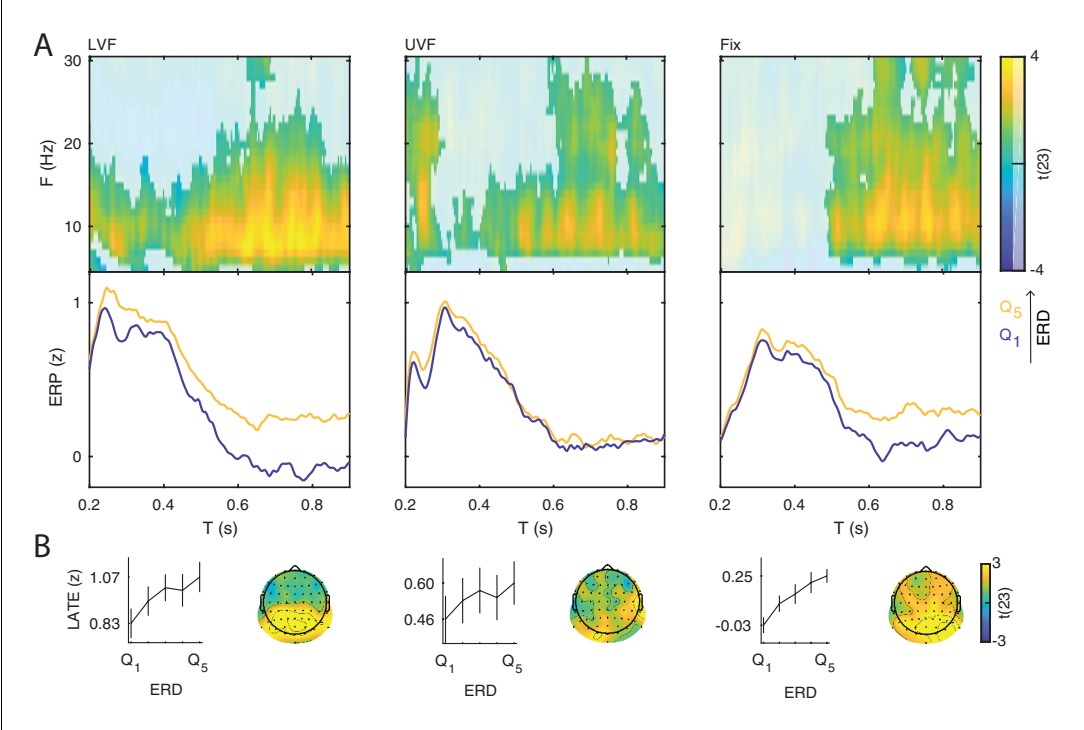

**Figure 5.** Interaction between the event-related desynchronization (ERD) and the late ERP component. A positive relationship between the ERD magnitude and the late ERP amplitude is another prerequisite for the baseline-shift mechanism. (**A**) Group-level t-statistics maps of the difference in late ERP amplitude (>0.200 s) between states of weak ($Q_1$) and strong ($Q_5$) event-related desynchronization (ERD) on lower visual field (left panel, LVF), upper visual field (middle panel, UVF), and fixation-only trials (right panel, Fix). Positive values (yellow) indicate that the amplitude of the late ERP increases during states of strong ERD. The maps are averaged across electrodes of the significant cluster, and masked by a final alpha of 0.05 using two-sided cluster permutation testing. Note that the x-axis refers to post-stimulus ERP time, while the y-axis refers to the ERD frequency. Bottom insets: visualization of the normalized late ERP time course separately for states of strong ($Q_5$, yellow) and weak ($Q_1$, blue) ERD, computed at the electrode and frequency of most positive t-statistics. Time 0 indicates stimulus/fixation-target onset. (**B**) Group-average normalized ERP amplitude on trials sorted from weak ($Q_1$) to strong ($Q_5$) ERD, calculated at the ERP time point and ERD electrode and frequency of most positive t-statistics. The amplitude of the late ERP increases linearly as a function of ERD. Error bars indicate ± SEM. The topographies show the positive t-statistics of the significant clusters at the time point and frequency of most positive t-statistics for each trial type. Black dots represent electrodes comprising the significant positive cluster. Taken together, these results show that states of strong ERD are associated with an enhancement of the late ERP component, consistent with the baseline-shift account.

DOI: https://doi.org/10.7554/eLife.43620.008

*De Blasio and Barry, 2013*; *Roberts et al., 2014*). However, the results have been mixed: several studies found decreased ERP amplitude during states of strong prestimulus power (*Rahn and Başar, 1993*; *Roberts et al., 2014*; *Becker et al., 2008*; *Başar and Stampfer, 1985*; *Jasiukaitis and Hakerem, 1988*) while others studies found increased ERP amplitude during states of strong prestimulus power (*Dockree et al., 2007*; *Becker et al., 2008*; *Roberts et al., 2014*; *Başar and Stampfer, 1985*; *Jasiukaitis and Hakerem, 1988*; *Barry et al., 2000*). Therefore, the precise mechanism by which prestimulus oscillations modulate sensory responses constitutes a continuing subject of debate in neuroscience. We addressed this issue by considering different potential mechanisms of ERP generation and how they may depend on prestimulus oscillations: namely, the additive and baseline-shift mechanisms. First, early ERP components are thought to reflect neural activation in sensory areas adding to prestimulus activity. Accordingly, we predicted that early ERP components would be attenuated during states of strong prestimulus low-frequency power, as suggested by physiological inhibition accounts (*Haegens et al., 2011*). Second, the late ERP component is likely generated by a post-stimulus modulation of ongoing oscillations (ERD) via baseline shift (*Nikulin et al., 2007*). Accordingly, we predicted that the late ERP component would be enhanced during states of strong prestimulus power, which are also associated with strong ERD. These predictions were confirmed by the data.

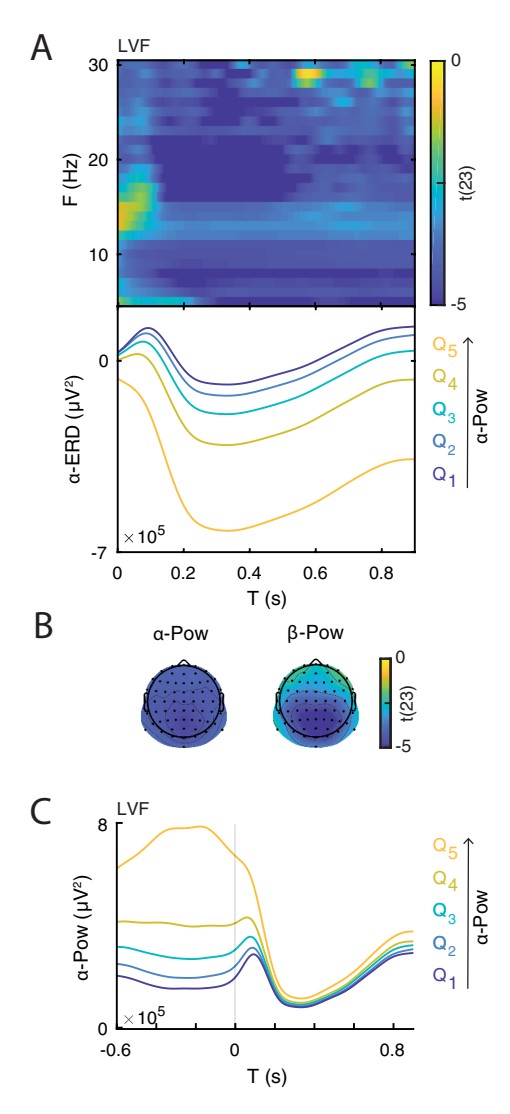

**Figure 6.** Illustration of the relationship between event-related desynchronization (ERD) and prestimulus power. (**A**) Group-level t-statistics map of the difference in ERD magnitude between states of weak ($Q_1$) and strong ($Q_5$) prestimulus power on lower visual field (LVF) trials. Negative values (blue) indicate that the ERD magnitude increases during states of strong prestimulus power. The map is averaged across occipital electrodes. Because of the circularity in the computation of prestimulus power and ERD, the t-values are inflated and only shown for illustrative purposes. No significance testing was run for this analysis. Note that the x-axis refers to post-stimulus ERD time, while the y-axis refers to the frequency of prestimulus oscillatory power. Bottom insets: visualization of the normalized ERD time course separately for the five bins of prestimulus power (weak to strong: $Q_{1-5}$, blue to yellow), computed at the electrode and frequency of most negative t-statistics. Time 0 indicates stimulus onset. (**B**) The topographies

*Figure 6 continued on next page*

## Functional inhibition mechanism

The results on the early ERP components in this study (<0.200 s: C1/N150) confirm and extend findings from past literature in the visual and auditory modalities. Specifically, previous studies in the visual modality found a negative relationship between prestimulus $\alpha$-band power and the amplitude of the visual N1P2 (i.e., amplitude difference between N1 and P2 components: *Rahn and Başar, 1993*), N1 (*Roberts et al., 2014*) and N175 components (*Becker et al., 2008*). A similar pattern of results was found for the N100 in the auditory modality (*Başar and Stampfer, 1985*; *Jasiukaitis and Hakerem, 1988*). It is important to note that previous results (e.g., *Başar and Stampfer, 1985*; *Becker et al., 2008*) that have been used to support the functional inhibition account could actually have been caused by a baseline shift. In the current study, we leverage the fact that the C1 has a well-known polarity reversal as a function of the visual field of the stimulus. By showing that the absolute amplitude of the C1 component is diminished by stronger prestimulus power, regardless of polarity, we can rule out a baseline shift which would affect both polarities in the same direction (e.g, a net increase or decrease of voltage). Additionally, previous results may have also have been due to (1) fluctuations of the 1/f aperiodic signal (which affect total-band power estimates, see *Appendix 1—figure 1*), or (2) fluctuations of sleepiness (which affect both oscillatory power and ERP amplitude, see *Appendix 1—figure 2*). In the current study, we confirmed that the early ERP amplitude was indeed reduced by oscillatory power, rather than just the 1/f aperiodic signal, and that this effect was not an epiphenomenon due to sleepiness. Taken together, these findings provide the first conclusive evidence for the functional inhibition effect of prestimulus oscillations on the early ERP amplitude.

Unlike in the visual and auditory modality, the relationship between prestimulus power and early ERP components in the somatosensory modality (e.g., N1) may be non-linear (inverted U-shaped: *Zhang and Ding, 2010*; *Ai and Ro, 2014*; *Forschack et al., 2017*) or vary across early components (i.e., negative for M50 and positive for M70, P35 and P60: *Jones et al., 2009*; *Nikouline et al., 2000*). Similarly, the relationship between prestimulus power and somatosensory perceptual performance has been found to have an inverted U-shape (*Linkenkaer-Hansen et al., 2004*), or to be linear (*Haegens et al., 2011*;

*Figure 6 continued*

show the negative t-statistics averaged for the $\alpha$- (8–12 Hz) and $\beta$-band (13–30 Hz) and for the late time window (0.200–0.900 s) on LVF trials. (C) Group-average power envelope shown separately for the five bins of prestimulus power (weak to strong: $Q_{1-5}$, blue to yellow) averaged for the $\alpha$-frequency band and occipital electrodes on LVF trials. Note that poststimulus power is similar across different prestimulus $\alpha$-band bins. These results show that states of strong prestimulus power are followed by strong ERD, consistent with the baseline-shift account.
DOI: https://doi.org/10.7554/eLife.43620.009

*Craddock et al., 2017*). Taken together, these findings suggest that in the somatosensory domain distinct functional mechanisms may map onto low-frequency oscillations.

Importantly, several studies report a positive relationship between prestimulus $\alpha$-band power and the amplitude of the visual N100 (*Jansen and Brandt, 1991*; *Brandt, 1997*), N1P2 (*Brandt et al., 1991*; *Brandt and Jansen, 1991*; *Barry et al., 2000*) and P200 (*Jansen and Brandt, 1991*). Thus, these studies appear inconsistent with the current results and other studies in the visual and auditory modality. However, a direct comparison is difficult for several reasons. First, some of these studies delivered visual stimuli to participants with eyes closed (*Brandt and Jansen, 1991*; *Brandt and Jansen, 1991*; *Brandt, 1997*; *Jansen and Brandt, 1991*). Instead, the majority of previous studies (including ours) delivered visual stimuli to participants with eyes open. It is known that oscillatory power in low frequencies has different spectral (*Barry et al., 2007*) and functional (*Kaida et al., 2006*) properties depending on whether subjects' eyes are open or closed; thus, these inconsistencies may be due to the eyes-open/closed difference. Second, unlike our study, which analyzed a broad frequency band, 64 electrodes, and an extensive post-stimulus time window (0–0.900 s), most previous studies only analyzed a narrow frequency band, few electrodes and a single time point. Therefore, it is possible that the inconsistent effects in previous studies were due to this selective (and under-sampled) analysis of EEG data. Third, some previous studies lack: (1) sufficient description of the EEG analysis (e.g., *Brandt, 1997*), (2) adequate statistical power (due to low number of participants or trials: e.g *Brandt and Jansen, 1991*; *Brandt and Jansen, 1991*; *Brandt, 1997*), and (3) quantitative statistical testing (*Brandt, 1997*). Consequently, this makes it difficult to compare these studies to the current one.

The present results have implications for the role of low-frequency oscillations in perceptual decision-making and in the top-down control over sensory processing (e.g., by spatial attention). In fact, numerous studies have found that weak prestimulus $\alpha$-band power increases observers' hit rates for near-threshold stimuli (*Ergenoglu et al., 2004*; *Zhang and Ding, 2010*; *Chaumon and Busch, 2014*). More recently, studies have demonstrated that this effect is not due to more accurate perception, but rather to a more liberal detection bias (*Limbach and Corballis, 2016*; *Iemi et al., 2017*; *Craddock et al., 2017*; *Iemi and Busch, 2018*) and a concomitant increase in confidence (*Samaha et al., 2017b*) and subjective visibility (*Benwell et al., 2017*). Unfortunately, conventional signal detection theory cannot be used to distinguish between alternative kinds of bias (*Morgan et al., 2013*; *Witt et al., 2015*). Specifically, a change in bias could be due to a change in the observer's deliberate decision strategy without any change in sensory processing (decision bias). Alternatively, a change in bias could be due to a change in the subjective appearance of stimuli (perceptual bias): liberal perceptual bias during states of weak prestimulus power could result from increased neural excitability amplifying both neural responses to sensory stimuli (thereby increasing hit rates) and responses to noise (thereby increasing false alarm rates). Interestingly, the present finding that the C1 is amplified during states of weak prestimulus power, indicates that even the earliest visual evoked responses are modulated by prestimulus oscillations. Even though we could not study an equivalent amplification of responses to noise using the present paradigm, this finding supports a perceptual bias mechanism more than a decision bias mechanism.

Furthermore, many experiments have noted a relationship between the topography of $\alpha$-band power and the focus of covert spatial attention (e.g., *Samaha et al., 2016*). However, considerable debate exists as to whether this preparatory $\alpha$-band modulation (and hence spatial attention) is capable of modulating feed-forward visual input (e.g., the C1 component). Our results show a clear impact of spontaneous $\alpha$ and $\beta$-band power on C1 amplitudes, supporting the idea that attention-related low-frequency modulation can affect the earliest stages of sensory processing. However, it is possible that attention-related and spontaneous oscillations have different effects on the amplitude

of the C1 component. This question is a candidate for future investigation, ideally by using stimuli such as those employed here, which generated robust C1 responses.

## Baseline-shift mechanism

In this study we demonstrated that the late component of the visual ERP was generated by a modulation of non-zero mean oscillations via baseline shift. There are four requirements to demonstrate the baseline-shift mechanism. First, the ongoing oscillations must have a non-zero mean. To this end, we estimated the non-zero-mean property of resting-state oscillations using $AFAI$ and $BSI$. This analysis revealed that $\alpha$- and $\beta$-band oscillations were characterized by a negative non-zero-mean. The frequencies and electrodes of the significant cluster for $AFAI$ were more extended relative to the cluster for $BSI$. This could be due to the fact that, unlike $BSI$, $AFAI$ is biased by harmonics and thus it reflects both non-zero mean oscillations and the 'comb-shape' of oscillations, which may yield amplitude asymmetries even when the signal has a zero mean (*Nikulin et al., 2010a*; *Nikulin et al., 2010b*). Thus, $AFAI$ is expected to be susceptible to more asymmetry-related features with larger spatial and spectral distribution compared to $BSI$.

Second, sensory stimuli must modulate the amplitude of ongoing oscillations. To test this requirement, we estimated the power modulation in the post-stimulus window relative to a prestimulus baseline (i.e., event-related oscillations: $ERD/ERS$). We observed a strong ERD in frequencies between 6 and 30 Hz in all three trial types. On Fix trials there were no robust early evoked components due to the lack of strong visual input, yet we observed an ERD following the same spatio-temporal dynamics as on stimulation trials (though of a lesser magnitude). In addition to the ERD, we also observed a strong ERS below 8 Hz on stimulation trials (but not on Fix trials) possibly reflecting a leakage from the robust evoked components measured during the early time window.

Third, the non-zero mean and the late ERP must have opposite polarity in case of ERD. Consistent with this requirement, our results showed that oscillations with a negative non-zero mean were associated with a late ERP component of positive polarity.

Fourth, ERD magnitude must correlate with the amplitude of the late ERP component. Our results indicated that strong ERD of non-zero mean oscillations was associated with enhanced ERP amplitude during the late time window. Importantly, the late ERP component was characterized by a topography and time-course similar to the ones of the ERD, consistent with *Mazaheri and Jensen (2008)*. In sum, these findings confirm the four requirements necessary to demonstrate the baseline-shift mechanism for the generation of the late ERP component.

The baseline-shift account predicts that stronger ERD occurs during states of stronger prestimulus power, which generates a greater baseline shift. In the case of negative non-zero-mean oscillations, this process results in an enhancement of the late ERP component with positive polarity. To test this prediction, we analyzed how prestimulus power is related to the ERD magnitude, and in turn to the amplitude of the late ERP component. We found that trials with strong prestimulus power were related to strong ERD magnitude, consistent with previous studies (*Min et al., 2007*; *Becker et al., 2008*; *Tenke et al., 2015*; *Benwell et al., 2018*). Due to circularity in these measures (i.e., ERD is computed with prestimulus power), the statistical estimates of this relationship are inflated. However, this pattern of results corroborates the prediction of the baseline-shift account. Specifically, we found that $\alpha$-band power is reduced to approximately the same level regardless of prestimulus power (*Figure 6C*). Accordingly, whereas the average prestimulus voltage is expected to differ between different prestimulus power bins due to the non-zero-mean property of neural oscillations, the average post-stimulus voltage in the late window is expected to be the same regardless of prestimulus power (*Figure 1B*, upper panel). The baseline-shift account predicts that subtracting a stronger prestimulus signal (strong power bin) yields a stronger shift of the EEG signal from the prestimulus baseline, and thus a stronger late ERP component (*Figure 1B*, lower panel). Consistent with our prediction, we also found a positive relationship between prestimulus power and the amplitude of the late ERP component.

These results confirm and extend previous findings in visual and auditory modalities. Specifically, in the visual modality prestimulus $\alpha$-band power was found to be positively correlated with the ERP amplitude in a late time window starting from 0.200 s relative to stimulus onset (0.550–0.800 s: *Dockree et al., 2007*; 0.220–0.310 s: *Becker et al., 2008*; 0.400 s: *Roberts et al., 2014*). A similar pattern of results was found on late ERP components in the auditory modality (0.250–0.800 s: *Jasiukaitis and Hakerem, 1988*; 0.400 s: *Başar and Stampfer, 1985*; 0.200–0.500 s: *Barry et al.,*

*2000*). Previous studies (e.g., *Barry et al., 2000*) were unable to explain the positive relationship between $\alpha$-power and ERP amplitude, which appeared inconsistent with the functional inhibition account (*Haegens et al., 2011*). Therefore, this study resolves this apparent inconsistency in previous literature, by demonstrating that this positive relationship can be accounted for by the baseline-shift mechanism, rather than functional inhibition.

It may seem surprising that the effects of prestimulus power on the late ERP occurred after the peak of the classically-defined slow component at approximately 0.300 s relative to stimulus onset (*Nikulin et al., 2007*; *Mazaheri and Jensen, 2008*). While early ERP components are likely generated primarily through the additive mechanism (because ERD is negligible in this time window), late ERP components can have a contribution from both additive and baseline-shift mechanisms. Functional inhibition of additive components in the initial part of the late time window might have canceled the amplification effect due to the baseline shift. This cancellation might explain the lack of a significant effect at the peak of the late component. In contrast, the ERP during the later time window (>0.400 s) is more likely to show primarily baseline-shift-generated components and thus is more susceptible to the amplification effect of prestimulus power.

We conclude that the positive modulation of the late ERP component is directly produced by the modulation of ERD magnitude as a function of prestimulus power. It is important to note that previous results on the late ERP component (e.g., *Dockree et al., 2007*; *Becker et al., 2008*) may have been influenced by (1) fluctuations of the 1/f aperiodic signal (which affect total-band power estimates, see *Appendix 1—figure 1*), or (2) fluctuations of sleepiness (which affect both oscillatory power and ERP amplitude, see *Appendix 1—figure 2*). In the current study, we confirmed that the late ERP amplitude was indeed amplified by oscillatory power, rather than just the 1/f aperiodic signal, and that this effect was not an epiphenomenon due to sleepiness. This provides the first evidence that the effect of prestimulus oscillations on the late ERP component is due to the mechanism of baseline shift.

## Comparison between functional inhibition and baseline shift

It is important to highlight that, while the functional inhibition account describes a (proposed) physiological mechanism, the baseline-shift account describes an effect that is largely the consequence of specific properties of the signal and the way we analyze it. That is, baseline shift is a result of a combination of preconditions including signal properties (non-zero mean), the occurrence of an ERD, as well as conventional signal processing procedures (i.e., baseline correction). The modulation of late responses predicted by a baseline shift can only exist if these preconditions are met, while the functional inhibition account generalizes to cases involving zero-mean oscillations, does not depend on the presence of an ERD, and can be established using different measures of brain activity (i.e., not limited to ERPs).

## $\alpha$- and $\beta$-band oscillations: a similar functional role?

The results of this study demonstrate a modulatory role of low-frequency oscillations on ERP amplitude. Both effects of prestimulus oscillations on early and late ERP components were characterized by a broad frequency range spanning the $\alpha$- and $\beta$-band. Likewise, the ERD and the non-zero-mean property of oscillations were found for both the $\alpha$-band and $\beta$-band. Specifically, $\alpha$-band ERD was sustained in time while $\beta$-band ERD was more transient, consistent with previous studies (e.g., *Salenius et al., 1997*). This suggests that $\beta$-band ERD may also reflect the generation of the late ERP component. One possible explanation for this multi-band effect can be the non-sinusoidal nature of neural oscillations (e.g., 'comb-shape': *Cole and Voytek, 2017*), which applies to both $\alpha$- and $\beta$-bands. In this case the event-related power modulation would similarly affect $\alpha$- and $\beta$-band activity. Because of such comodulation, baseline-shifts associated with $\alpha$-band oscillations would also appear for $\beta$-band oscillations, resulting in similar *BSI* and *AFAI* for both frequency bands (*Nikulin et al., 2010a*).

The $\beta$-band effect may seem surprising given that the majority of past literature focused solely on the $\alpha$-band due to its high signal-to-noise ratio compared to other frequencies. However, the broad frequency range of the effects reported in this study is in line with studies reporting a temporal and spatial co-modulation of $\alpha$- and $\beta$-band oscillations (*Bastos et al., 2015*; *Lakatos et al., 2016*; *Michalareas et al., 2016*). It is also consistent with recent studies reporting a similar relationship

between $\alpha$- and $\beta$-band prestimulus power, perceptual reports (*Benwell et al., 2017*; *Iemi et al., 2017*; *Samaha et al., 2017a*; *Samaha et al., 2017b*; *Iemi and Busch, 2018*) and firing rate (*Watson et al., 2018*). Accordingly, it has been proposed that $\beta$-band oscillations exert an inhibitory function, similar to $\alpha$-band oscillations (*Spitzer and Haegens, 2017*; *Shin et al., 2017*; *Kilavik et al., 2013*).

## What about phase reset?

In this study we considered the additive (*Bijma et al., 2003*; *Shah et al., 2004*; *Mäkinen et al., 2005*; *Mazaheri and Jensen, 2006*) and baseline-shift (*Nikulin et al., 2007*; *Mazaheri and Jensen, 2008*) mechanisms for the generation of early and late ERP components (*Bijma et al., 2003*; *Shah et al., 2004*; *Mäkinen et al., 2005*; *Mazaheri and Jensen, 2006*), respectively. In addition to these mechanisms, some studies have proposed that the ERP can be generated by a reorganization of ongoing oscillations via phase reset (*Sayers et al., 1974*; *Makeig et al., 2002*; *Klimesch et al., 2004*; *Gruber et al., 2005*; *Fell et al., 2004*; *Fuentemilla et al., 2006*; *Hanslmayr et al., 2007*; *Sauseng et al., 2007*). According to the phase-reset account, the phases of ongoing oscillations are aligned (i.e., phase-reset) by the stimulus; as a consequence, averaging these phase-locked oscillations across trials does not lead to phase cancellation in the post-stimulus window, resulting in the generation of ERP components. Specifically, the phase reset of an oscillation at a particular frequency is expected to generate a component with similar frequency characteristics: for example, the $\alpha$-band phase reset is thought to generate early ERP components of the same polarity with an inter-peak latency at approximately 100 ms (as the C1 and N150 on UVF trials, see *Figure 2B*, middle panel) (*Hanslmayr et al., 2007*; *Sauseng et al., 2007*). Several studies therefore proposed that ERP components at different latencies (with different frequency characteristics) are generated by a phase reset of $\alpha$- and $\beta$-band oscillations (P1 and N1: *Klimesch et al., 2004*; *Makeig et al., 2002*; *Gruber et al., 2005*) or $\delta$- and $\theta$-band oscillations (P300: *Fell et al., 2004*). However, please note that, while phase reset of $\alpha$-band oscillations may explain the generation of the early ERP components with positive polarity on UVF trials, it cannot account for the opposite polarity of the C1 component on LVF trials (see *Figure 2B*).

Although our experiment was not designed to test the phase-reset hypothesis, we see two possible predictions that a phase-reset account could make on the relationship between prestimulus oscillatory power and ERP amplitude (*Hanslmayr et al., 2007*; *Sauseng et al., 2007*). On the one hand, it has been argued that phase reset in response to a stimulus can only occur if the oscillation already exists prior to the reset (i.e., during the prestimulus window). It follows that any ERP component generated by phase reset is likely to be absent during desynchronized states (i.e., weakest power bin) (*Shah et al., 2004*; *Sauseng et al., 2007*). This suggests that trials with weakest prestimulus power may result in less prominent phase reset, which would manifest as a reduction of the ERP amplitude. Accordingly, we would expect a positive relationship between prestimulus oscillations in the $\alpha$ and $\beta$ bands and the amplitude of the C1 and N150 components on UVF trials, which occur with an inter-peak latency of approximately 70–80 ms. Contrary to this prediction, our study showed that the C1 and N150 amplitudes on UVF trials were negatively correlated with prestimulus $\alpha$- and $\beta$-band oscillations. In addition, we found that the amplitude of the slow ERP component was positively correlated with prestimulus $\alpha$- and $\beta$-band oscillations. However, this correlation is unlikely to be accounted for by phase reset of $\alpha$- and $\beta$-band oscillations because these rhythms are much faster than the one reflected in the slow ERP component (i.e., $\delta$ rhythm).

On the other hand, one can argue that strong oscillations represent a state with pronounced neuronal synchronization that is not easily affected by weak sensory inputs, as also shown in previous modeling work (*Hansel and Sompolinsky, 1996*). Thus, during states of strong ongoing oscillations, phase-reset may be harder to be achieved and, consequently, is unlikely to result in ERP generation. Accordingly, this predicts an ERP attenuation by prestimulus power, consistent with our results during the early time window. It is worth noting that, in this study, it is difficult to distinguish whether this attenuation affects ERP components generated by additive or phase-reset mechanisms; invasive electrophysiological recordings allowing for higher spatial resolution might be required to address this particular question (*Hanslmayr et al., 2007*; *Telenczuk et al., 2010*). Regardless of the underlying mechanisms of ERP generation, our results during the early time window can be explained by the functional inhibition account.

## Conclusion

This study demonstrates that spontaneous fluctuations of oscillatory brain activity modulate the amplitude of visual ERP via two distinct mechanisms: (1) functional inhibition of the early additive ERP components and (2) baseline shift affecting the late ERP component. Therefore, these findings show that neural oscillations have concurrent opposing effects on ERP generation. Distinguishing between these effects is crucial for understanding how neural oscillations control the processing of incoming sensory information in the brain.

# Materials and methods

## Participants

Previous studies on the relationship between neural oscillations and ERPs have typically reported samples of 7–19 participants (e.g., *Jasiukaitis and Hakerem, 1988*; *Brandt and Jansen, 1991*; *Rahn and Başar, 1993*; *Nikulin et al., 2007*; *Mazaheri and Jensen, 2008*; *Becker et al., 2008*; *van Dijk et al., 2010*; *Rajagovindan and Ding, 2011*). To ensure a robust estimate of our neuro-physiological effect and account for potential missing data (e.g., due to artifacts), we recruited a larger sample of 27 participants (mean age: 26.33, SEM = 0.616; 14 females; three left-handed). All participants had normal or corrected-to-normal vision and no history of neurological disorders. Prior to the experiment, written informed consent was obtained from all participants. All experimental procedures were approved by the ethics committee of the German Psychological Society. Two participants were excluded before EEG preprocessing because of excessive artifacts. One participant was excluded after preprocessing because no C1 component could be detected, unlike the rest of the sample. A total of 24 participants were included in the analysis.

## Stimuli and experimental design

The experiment was written in MATLAB (RRID:SCR_001622) using the Psychophysics toolbox 3 (RRID:SCR_002881: *Brainard, 1997*; *Pelli, 1997*). The experiment included a resting-state session and a stimulation session, lasting approximately 1.5 hr including self-paced breaks.

The resting-state session was divided in two recording blocks, each of which lasted 5.5 min, separated by a short self-paced break. In this session participants were required to keep their eyes open and fixate on a mark located at the center of the screen, to avoid movements and not to think of anything in particular.

In the stimulation session, participants were presented with visual stimuli specifically designed to elicit a robust C1 component of the visual ERP. The C1 is described as the earliest component of the visual ERP with a peak latency between 0.055 and 0.09 s and an occipital topography. The C1 component is regarded as an index of initial afferent activity in primary visual cortex, because of its early latency and polarity reversal with reference to V1 anatomy (*Di Russo et al., 2002*; *Di Russo et al., 2003*).

The stimuli consisted of full-contrast bilateral black-and-white checkerboard wedges. Each wedge corresponded to a radial segment of an annular checkerboard (spatial frequency = 5 cycles per degree) with inner and outer circle of 3 and 10° of eccentricity relative to a central fixation point, respectively. Each wedge covered 3.125% of the area of the annular checkerboard and spanned 11.25° of visual angle (*Vanegas et al., 2013*).

In each stimulation trial, a pair of wedges was presented bilaterally either in the UVF or LVF with equal probability. UVF and LVF stimulus positions were located at polar angles of 25° above and 45° below the horizontal meridian, respectively. These asymmetrical positions for UVF and LVF stimuli ensure a stimulation of primarily lower and upper banks of the calcarine fissure, respectively (*Aine et al., 1996*; *Clark et al., 1994*; *Di Russo et al., 2002*), resulting in a polarity reversal of scalp potentials. A positive C1 component is obtained by LVF stimulation, while a negative C1 component is obtained by UVF stimulation (*Di Russo et al., 2002*; *Di Russo et al., 2003*).

The stimuli were presented for a duration of 0.100 s (*Fu et al., 2010*; *Ding et al., 2014*; *Kelly et al., 2008*) at full contrast (*Hansen et al., 2016*; *Vanegas et al., 2013*) on a gray background that was isoluminant relative to the stimuli's mean luminance. The stimuli were presented at a viewing distance of 52 cm, on a cathode ray tube monitor operated at 100 Hz, situated in a dark, radio-

frequency-interference shielded room. Throughout the experiment, fixation distance and head alignment were held constant using a chin rest.

For each participant the stimulation session included 810 trials, divided into nine recording blocks. In each block, 60 trials contained stimuli in either LVF or UVF with equal probability (stimulation trials), while 30 trials were stimulus-absent (fixation-only trials). Trial type and stimulation field were randomized across trials within each block. To ensure that the participants maintained the gaze to the center, we included a discrimination task at the central fixation mark, similar to previous studies (*Di Russo et al., 2002*; *Chen et al., 2016*). On stimulation trials the central fixation mark turned into either one of two equally probable targets ('>' or '<') during stimulus presentation for a duration of 0.100 s. On Fix trials, the change of the central fixation mark into the target occurred during a 0.100 s window between 1.8 and 2.4 s relative to trial onset. Note that, while the targets might have caused an involuntary shift of lateral attention, these effects would have cancelled out across trials because of the fully randomized presentation of the targets (each recording block included 50% '<' targets and 50% '>' targets). Discrimination performance at or close to ceiling (i.e., 100% correct responses) was expected if gaze was maintained on the central fixation mark. This task also ensured that the participants remained alert throughout the experiment. Mean accuracy in the fixation task was 94.85% (SEM = 0.0109) and did not significantly differ between trial types (p>0.05), indicating that participants were able to maintain central fixation. Incorrect trials were discarded from further analysis (mean = 41; SEM = 8.620). On average we analyzed 761 (SEM = 9.824) trials per participant.

After target offset, the fixation mark was restored for a duration of 0.100 s. After this delay, the fixation mark turned into a question mark, which instructed the participants to deliver a response via a button press with their dominant hand. After the button press, the fixation mark was displayed again and a new trial started. The following stimulus presentation or fixation task occurred after a variable delay chosen from a uniform distribution between 1.8 and 2.4 s.

In addition to the fixation task, to further prevent eye movements, all participants were trained prior to EEG recording to maintain fixation on the central mark and their fixation ability was monitored throughout the experiment using the electro-oculogram (EOG). Moreover, we used a shape of the fixation mark specifically designed to maximize stable fixation (*Thaler et al., 2013*).

To control for an effect of sleepiness on the level of ongoing low-frequency power, we asked participants to report their level of sleepiness at the end of each block during resting-state and stimulation session. We used the Karolinska Sleepiness Scale (KSS), which has been validated as an indicator of objective sleepiness (*Kaida et al., 2006*). The KSS scale consists of a nine-point Likert-type scale ranging from 1 (extremely alert) to 9 (very sleepy) that represents the sleepiness level during the immediately preceding 5 minutes. The scale was presented on the screen at the end of every block and participants were instructed to report how alert they felt during the immediately preceding block by pressing the corresponding number on the keyboard (1–9). After the button press, participants could take a self-paced break and the following block was initiated via button-press.

## EEG recording and preprocessing

EEG was recorded with a 64-channel Biosemi ActiveTwo system at a sampling rate of 1024 Hz. Electrodes were placed according to the international 10–10 system (electrode locations can be found on the Biosemi website: https://www.biosemi.com/download/Cap_coords_all.xls). The horizontal and vertical electro-oculograms were recorded by additional electrodes at the lateral canthi of both eyes and below the eyes, respectively. As per the BioSemi system design, the Common Mode Sense (CMS) and Driven Right Leg (DRL) electrodes served as the ground. All scalp electrodes were referenced online to the CMS-DRL ground electrodes during recording. Electrode impedance was kept below 20 mV. The raw data was recorded with ActiView (version 6.05).

The EEGLAB toolbox version 13, running on MATLAB (R2017b), was used to process and analyze the data (*Delorme and Makeig, 2004*). In both resting-state and stimulation sessions, the data were re-referenced to the mastoids and down-sampled to 256 Hz. In the resting-state session, data were epoched from 0 to 330 s relative to the start of the recording block. In the stimulation session, data were epoched from −1.6 to 1.3 s relative to the onset of the stimulus presentation on stimulation trials or to the onset of the fixation task on Fix trials. In both sessions, the data were then filtered using an acausal band-pass filter between 0.25 and 50 Hz. We manually removed gross artifacts such as eye blinks and noisy data segments. In the stimulation session, we discarded entire trials when a

blink occurred within a critical 0.500 s time window preceding stimulus/fixation-target onset, to ensure that participants' eyes were open at stimulus onset. Furthermore, we manually selected noisy channels on a trial-by-trial basis for spherical spline interpolation (*Perrin et al., 1989*). We interpolated on average 8 channels (SEM = 0.96) in 35 trials (SEM = 6.71). No channels were interpolated in the resting-state session. In both sessions, we transformed the EEG data using independent component analysis (ICA), and then we used SASICA (Semi-Automated Selection of Independent Components of the electroencephalogram for Artifact correction) (*Chaumon et al., 2015*) to guide the exclusion of IC related to noisy channels and muscular contractions, as well as blinks and eye movements. On average, we removed 7.9 (SEM 0.46) and 7.8 (SEM 0.60) out of 72 ICs in the resting-state and stimulation session, respectively.

## Event-related potentials

The aim of this study was to examine the influence of prestimulus oscillatory power on ERP amplitude. We used visual stimuli to specifically elicit a robust C1 component of the visual ERP, which reflects initial afferent activity of the primary visual cortex (*Di Russo et al., 2002*). For each participant we identified the electrode and time point with peak activity between 0.055 and 0.090 s after stimulus onset (peak C1 activity: *Di Russo et al., 2002*; *Bao et al., 2010*), separately for LVF and UVF trials. On Fix trials, no C1 component of the visual ERP is expected. To quantify the ERP for this trial type and to enable comparison with the stimulation trials, we averaged the EEG data across the subject-specific electrodes with peak C1-activity in the stimulation trials. We baseline corrected single-trial ERP estimates by subtracting the prestimulus signal baseline averaged across a 0.500 s prestimulus window.

## Event-related oscillations

We used time-frequency analysis to obtain a measure of ongoing oscillatory power and to estimate event-related oscillations (ERO). We first computed the stimulus-evoked, phase-locked activity (ERP) by averaging the EEG signal across trials. Then, we subtracted the average ERP from single-trial EEG signal. We applied this procedure separately for LVF, UVF, and Fix trials. This procedure ensures that the resulting ERO estimate does not contain stimulus-evoked activity (*Kalcher and Pfurtscheller, 1995*). Next, we applied a wavelet transform (Morlet wavelets, 26 frequencies, frequency range: 5–30 Hz, number of cycles increasing linearly from 3 to 8, time window: −1–1.3 s relative to stimulus onset) to the EEG signal. This procedure was performed separately for each electrode and trial type (LVF, UVF, and Fix) . We then quantified ERO as follows:

$$ERO = P_{post} - \mu(P_{pre}) \tag{1}$$

where $P_{post}$ is a the time course of post-stimulus oscillatory activity and $\mu(P_{pre})$ is the average ongoing power in a prestimulus window between −0.600 and −0.100 s relative to stimulus onset. This window for baseline correction was chosen based on *Mazaheri and Jensen (2008)* to circumvent the temporal smearing due to the wavelet convolution. ERO<0 indicates the presence of an event-related desynchronization (ERD), indicating stimulus-induced power attenuation. ERO>0 indicates the presence of an event-related synchronization (ERS), indicating stimulus-induced power enhancement. This procedure was performed separately for each frequency, electrode and participant.

## Hypothesis testing

In this study, we tested two potential mechanisms underlying the modulation of ERP amplitude by prestimulus power: namely, functional inhibition and baseline shift. Functional inhibition implies that the generation of early, additive ERP components is inhibited if stimulation occurs during a state of strong prestimulus activity. In other words, positive and negative early ERP components are expected to become less positive and less negative, respectively, during states of strong prestimulus power. The baseline-shift mechanism implies that states of strong prestimulus oscillations with a non-zero mean are followed by strong power suppression (ERD), which in turn results in an enhancement of the late ERP component.

## Prerequisites of the baseline-shift mechanism

Previous studies (*Nikulin et al., 2007*; *Mazaheri and Jensen, 2008*) proposed the following prerequisites for linking ERO to ERP generation (baseline-shift mechanism): (1) the ongoing oscillations must have a non-zero mean; (2) sensory stimuli must modulate ERO magnitude; (3) the non-zero mean and the late ERP component must have opposite polarity in case of ERD; (4) ERO magnitude is associated with the amplitude of the late ERP component.

## Estimation of non-zero-mean property of resting-state neural oscillations

The aim of the resting-state session was to estimate the non-zero-mean property of ongoing oscillations, which is known to be a critical requirement for generation of the late ERP via baseline shift (*Nikulin et al., 2007*; *Mazaheri and Jensen, 2008*). To this end, we used two analytical methods: namely, the Baseline Shift Index (*Nikulin et al., 2007*) and the Amplitude Fluctuation Asymmetry Index (*Mazaheri and Jensen, 2008*). In each participant, we estimated $BSI$ and $AFAI$ for resting-state oscillations for each electrode and frequency between 5 and 30 Hz.

Following *Nikulin et al. (2007)*, to quantify $BSI$ we first band-pass filtered the EEG signal using a $4^{th}$-order Butterworth filter centered at each frequency of interest ±1 Hz. Then, we extracted oscillatory power using the Hilbert transform. Additionally, we low-pass filtered the EEG signal using a $4^{th}$-order Butterworth filter with a 3 Hz cut-off frequency. The baseline shifts are low-frequency components, because the amplitude modulation of 8–30 Hz frequency oscillation can be detected only at frequencies considerably lower than 8 Hz. Thus, the low-frequency components are extracted by low-pass filtering the artifact-cleaned data at 3 Hz, based on previous studies (*Nikulin et al., 2007*; *Nikulin et al., 2010a*). We quantified the $BSI$ as the Spearman correlation coefficient ($\rho$) between the low-passed EEG signal and the band-passed power, separately at each frequency and electrode. Accordingly, $BSI = 0$ indicates no relationship between oscillatory power and low-passed signal, as expected for zero-mean oscillations. $BSI > 0$ indicates that strong oscillatory power is correlated with an increase of the low-passed signal, as expected for positive-mean oscillations; instead, $BSI < 0$ indicates that strong oscillatory power is correlated with a decrease of the low-passed signal, as expected for negative-mean oscillations.

The amplitude modulation of oscillations with a non-zero mean affects the amplitude of peaks and troughs differently. If the peaks are larger than the troughs relative to the zero line, the ERD will make the electric field go to zero and thus reduce the peaks more strongly than the troughs. It follows that, in this case, any amplitude modulation is expected to produce larger variance for peaks than troughs. The different modulation of peaks and troughs can be captured by $AFAI$. Following *Mazaheri and Jensen (2008)*, to quantify $AFAI$ we first band-pass filtered the EEG signal using a $4^{th}$-order Butterworth filter centered at each frequency of interest ±1 Hz, similarly to $BSI$ computation. Then, we identified the time points of peaks and troughs in the band-passed data. These time points were then used to obtain the signal values of peaks and troughs in the non-band-passed (broadband) signal. We quantified the $AFAI$ as the normalized difference between the variance of the peaks and troughs of the signal as follows:

$$AFAI = \frac{Var(S_p) - Var(S_t)}{Var(S_p) + Var(S_t)} \tag{2}$$

where $S_p$ and $S_t$ refer to the peak and trough values, respectively, estimated in the broadband signal, based on the band-passed signal at a specific frequency.

Accordingly, an $AFAI = 0$ indicates that the peaks and troughs are equally modulated (as for a signal that is symmetric relative to the zero line), as expected for zero-mean oscillations. An $AFAI \neq 0$ indicates amplitude asymmetry: namely, positive values indicate a stronger modulation of the peaks relative to the troughs (i.e., positive amplitude asymmetry or positive mean) and negative values indicate a stronger modulation of the troughs relative to the peaks (i.e., negative amplitude asymmetry or negative mean).

Note that the sign of the $BSI$ and $AFAI$ predicts the polarity of the late ERP component: specifically, if the sign is negative (oscillations with a negative mean) and positive (oscillations with a positive mean), event-related power suppression (ERD) will lead to a positive and negative deflection in the ERP, respectively (*Nikulin et al., 2007*; *Nikulin et al., 2010a*; *Mazaheri and Jensen, 2008*).

## Interaction between event-related potentials and oscillations

To provide evidence that the baseline-shift mechanism generates the late ERP component, we analyzed the relationship between ERP and ERO (ERS/ERD) across trials, as proposed by *Mazaheri and Jensen (2010)*. According to the baseline-shift account, states of strong ERD should result in an enhanced late ERP component. To this end, we first identified trials with particularly weak and strong ERO, and then tested how these trials differed in ERP amplitude during the late time window (>0.200 s). Specifically, we computed a trial-by-trial estimate of ERO magnitude at each electrode and frequency, averaged across the post-stimulus time window (0–0.900 s; see Event-related oscillations). We also computed a trial-by-trial estimate of the late ERP component at the subject-specific C1-peak electrode (>0.200 s; see Event-related potentials). We baseline corrected single-trial ERP estimates by subtracting the prestimulus signal baseline averaged across a 0.500 s prestimulus window. Then, for each frequency, and electrode, trials were sorted from weak to strong ERO, divided into five bins (*Linkenkaer-Hansen et al., 2004*; *Lange et al., 2012*; *Baumgarten et al., 2016*; *Iemi et al., 2017*), and the amplitude of the late ERP component was calculated for each bin. The binning was done separately for each trial type (LVF, UVF, and Fix) and participant. Furthermore, to enable a comparison of the late ERP component across bins in each participant, the number of trials in each bin was equated by removing the trials recorded at the end of the experiment. To test the hypothesis, we then compared the amplitude of the late ERP component between strongest and weakest ERO bins (see Statistical Testing for more details).

## Influence of prestimulus oscillations on event-related potentials and oscillations

We analyzed how prestimulus oscillatory activity influences ERP and ERO across trials. In this analysis, we identified trials with particularly weak and strong prestimulus oscillations, and then tested how these trials differed in the amplitude of the early and late ERP components and in the ERO magnitude. Specifically, we first computed a trial-by-trial estimate of oscillatory power with a Fast Fourier Transform (FFT) during a 0.500 s prestimulus window for each electrode and frequency. The FFT is advantageous here because, unlike wavelet convolution, the results of an FFT computed over the prestimulus period cannot be influenced by signals occurring in the post-stimulus window. We also computed a trial-by-trial estimate of ERP components during the early time window and ERP components during the late time window at the subject-specific electrode of C1 peak activity (see Event-related potentials). We baseline corrected single-trial ERP estimates by subtracting the ERP averaged across a 0.500 s prestimulus window. In addition, we computed a trial-by-trial estimate of the ERO (ERD/ERS) at the subject-specific electrode of C1 peak activity and at each frequency and time point in the post-stimulus time window (0–0.900 s; see Event-related oscillations). In this analysis, both ERP and ERO were quantified at the subject-specific electrode of C1 peak activity to enable comparison between the effects of prestimulus power on ERP and ERO. Then, for each frequency, and electrode, trials were sorted from weak to strong prestimulus power and divided into five bins (*Linkenkaer-Hansen et al., 2004*; *Lange et al., 2012*; *Baumgarten et al., 2016*; *Iemi et al., 2017*). For each bin we calculated the ERO magnitude and the amplitude of the early and late ERP components. The binning was done for each trial type and participant. Furthermore, to enable a comparison of ERO and ERP across bins in each participant, the number of trials in each bin was equated by removing the trials recorded at the end of the experiment. We then compared the ERO magnitude and the amplitude of the early and late ERP components between bins of strongest and weakest prestimulus power (see Statistical Testing for more details).

Because the estimates of prestimulus power were computed with an FFT, they reflect a mixture of periodic (i.e., oscillations) and aperiodic signals (i.e, 1/f 'background' noise: *Podvalny et al., 2015*; *Voytek et al., 2015*), referred to as total-band power. Therefore, we set out to determine whether ERP amplitude differed between bins of strongest and weakest periodic signal. To this end, we quantified a single-trial measure of the prestimulus power spectrum for the electrodes of maximal statistical effects separately for each component (C1, N150, and LATE), and for each trial type (LVF/UVF/Fix). Next, we parameterized the prestimulus power spectrum (1–48 Hz) into periodic and aperiodic signals (toolbox fooof: *Haller et al., 2018*; *Voytek et al., 2015*). First, we fitted the power spectrum with an aperiodic function defined by a slope and an offset. Then, to obtain a measure of the periodic signal, we subtracted this aperiodic function from the original power spectrum, resulting

in an aperiodic-adjusted power spectrum. Following the same procedure described above, we classified trials in five bins based on single-trial estimates of aperiodic-adjusted power at the frequencies of maximal statistical effects, and quantified the ERP amplitude (at the subject-specific channels of maximal C1 response and at the time points of maximal statistical effects) for each bin. Then, we compared the ERP amplitude between bins of strongest and weakest aperiodic-adjusted power (using paired-sample t-tests, corrected for multiple comparisons using FDR; *Benjamini and Hochberg, 1995*).

Because ongoing oscillatory activity (*Kaida et al., 2006*; *Zhang and Ding, 2010*; *Mathewson et al., 2009*; *Benwell et al., 2017*) and ERP amplitude (*Megela and Teyler, 1979*; *Budd et al., 1998*; *Truccolo et al., 2002*; *de Munck et al., 2004*) may co-vary over the course of an experiment as a function of time-varying variables such as sleepiness, their correlation could be epiphenomenal. To rule this out, we asked participants to report their level of sleepiness at the end of each experimental block using the KSS questionnaire (*Kaida et al., 2006*); see Stimuli and Experimental Design for more details). We then estimated how prestimulus power was related to KSS ratings throughout the stimulation session. Specifically, we computed a trial-by-trial estimate of prestimulus power for each electrode and frequency using an FFT on the 0.500 s prestimulus window. We obtained a trial-by-trial estimate of KSS scores by assigning each trial within a block with the KSS score collected at the end of the block. We then used Generalized Linear Modeling (GLM) to predict KSS ratings from prestimulus power at the single-trial level. For each participant, electrode and frequency, we fit a regression model of the following form:

$$KSS = \beta_0 + \beta_1 P + \varepsilon \tag{3}$$

where KSS is the subjective sleepiness ratings obtained with the KSS questionnaire, P the prestimulus power at each frequency and electrode, $\beta_1$ the estimated correlation coefficient indicating the contribution of P in explaining variability in KSS, and $\varepsilon$ the residual errors. To remove the sleepiness-related time-varying changes in ongoing power, we recomputed a trial-by-trial measure of prestimulus power as follows:

$$P_{KSSC} = P - \beta_1 KSS \tag{4}$$

where $P$ is the raw power estimates and $\beta_1$ the estimated GLM coefficient reflecting the sleepiness-power relationship. We then repeated the binning analysis on the early and late ERP amplitudes described above, with this new trial-by-trial estimate of power where sleepiness-related time-varying changes were ruled out ($P_{KSSC}$). If the relationship between prestimulus power and ERP is not determined by sleepiness affecting both variables, this new binning analysis would replicate the results of the analysis performed on raw power estimates.

## Statistical testing

In the resting-state session, within each subject, we first computed the *AFAI* and *BSI* at each frequency and electrode. For the group-level statistical inference, we then tested whether the *AFAI* and *BSI* were significantly different from 0 across the sample of participants.

In the stimulation session, within each subject, we first computed: (1) the difference in the late ERP component between the weakest and strongest ERO bins, (2) the difference in the ERD magnitude between the weakest and strongest prestimulus power bins, and (3) the difference in ERP between the weakest and strongest prestimulus power bins (separately for the early and late time window). For the group-level statistical inference, we computed the t-statistics of these differences (ΔV) against the null hypothesis that there was no difference between the bins.

No significance testing was run for the analysis of the relationship between ERO and prestimulus power due to the circularity of these measures. For all other analyses, a non-parametric cluster based permutation test was used to determine significant effects (*Maris and Oostenveld, 2007*). By clustering neighboring samples (i.e., based on temporal-spectral-spacial adjacency), that show the same effect, this test deals with the multiple comparison problem while taking into account the dependency of the data. We obtained a distribution of the variables of interest (i.e., *AFAI/BSI* for the resting-state session and ΔV for the stimulation session) under the null hypothesis by randomly permuting their signs 1000 times across participants. On each iteration, we tested the resulting variables with a two-tailed t-test against zero and computed the sum of the t-values within the largest

contiguous cluster of significant frequency-electrode (in the resting-state session) or time-frequency-electrode points (in the stimulation session) that exceeded an a priori threshold (cluster alpha = 0.05), resulting in a distribution of t-sums expected under the null hypothesis. A final p-value was calculated as the proportion of t-sums under the null hypothesis larger than the sum of t-values within clusters in the observed data. Because the cluster permutation test is based on sampling all time points, and the ERP signal comprises early, fast components and late, slow components having different statistical properties, we decided to test for significant effects separately during the early (<0.200 s) and late (>0.200 s) time windows. To correct for multiple comparisons due to running the test twice on the same time series, we divided the final permutation alpha by 2 (final alpha = 0.025, bonferroni corrected) and considered effects significant only if their p-values were below this threshold. We performed the statistical test separately for positive and negative clusters as recommended by *Maris and Oostenveld (2007)* for a two-sided cluster permutation test. We focused the statistical analysis on all electrodes, on frequencies from 5 to 30 Hz and between 0 and 0.900 s relative to stimulus onset.

## Acknowledgements

This work was supported by the HSE Basic Research Program and the Russian Academic Excellence Project '5–100' (LI, VVN), the Silvio O Conte Center for Active Sensing (P50 MH109429, LI), by a grant from the German Research Foundation (DFG) to NAB (BU2400/9-1), and by a grant from the Netherlands Organization for Scientific Research to SH (NWO 016.Vidi.185.137).

We thank Francesco Di Russo and Volodymyr B Bogdanov for help with designing the stimuli, Johanna Rehder for help with the literature search, and Charles Schroeder and Omri Raccah for comments on the manuscript.

## Additional information

### Competing interests

Saskia Haegens: Reviewing editor, *eLife*. The other authors declare that no competing interests exist.

### Funding

| Funder | Grant reference number | Author |
|---|---|---|
| Ministry of Education and Science of the Russian Federation | Russian Academic Excellence Project '5-100' | Luca Iemi Vadim V Nikulin |
| National Research University Higher School of Economics | Basic Research Program | Luca Iemi Vadim V Nikulin |
| Silvio O Conte Center for Active Sensing | P50 MH109429 | Luca Iemi |
| Deutsche Forschungsgemeinschaft | BU2400/9-1 | Niko A Busch |
| Nederlandse Organisatie voor Wetenschappelijk Onderzoek | NWO 016.Vidi.185.137 | Saskia Haegens |
| National Institute of Mental Health | MH095984 | Jason Samaha |

The funders had no role in study design, data collection and interpretation, or the decision to submit the work for publication.

### Author contributions

Luca Iemi, Conceptualization, Resources, Data curation, Software, Formal analysis, Validation, Investigation, Visualization, Methodology, Writing—original draft, Project administration, Writing—review and editing; Niko A Busch, Conceptualization, Supervision, Methodology, Writing—review

and editing; Annamaria Laudini, Conceptualization, Data curation, Software, Validation, Investigation, Writing—review and editing; Saskia Haegens, Supervision, Writing—review and editing; Jason Samaha, Conceptualization, Writing—review and editing; Arno Villringer, Funding acquisition, Writing—review and editing; Vadim V Nikulin, Conceptualization, Resources, Supervision, Funding acquisition, Methodology, Writing—review and editing

## Author ORCIDs

Luca Iemi ⓘ https://orcid.org/0000-0002-7178-2252
Saskia Haegens ⓘ https://orcid.org/0000-0002-9676-6275
Jason Samaha ⓘ http://orcid.org/0000-0001-8010-5993

## Ethics

Human subjects: Prior to the experiment, written informed consent was obtained from all participants. All experimental procedures were approved by the ethics committee of the German Psychological Society (reference number: Nb092013).

## Decision letter and Author response

Decision letter https://doi.org/10.7554/eLife.43620.017
Author response https://doi.org/10.7554/eLife.43620.018

## Additional files

### Supplementary files

• Transparent reporting form
DOI: https://doi.org/10.7554/eLife.43620.010

### Data availability

All data generated or analyzed during this study have been deposited to the Open Science Framework (https://osf.io/yn6gb).

The following dataset was generated:

| Author(s) | Year | Dataset title | Dataset URL | Database and Identifier |
|---|---|---|---|---|
| Iemi L, Niko A Busch, Annamaria Laudini, Saskia Haegens, Jason Samaha, Arno Villringer, Vadim V Nikulin | 2018 | Multiple mechanisms link prestimulus neural oscillations to sensory responses | https://osf.io/yn6gb | Open Science Framework, yn6gb |

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

## Appendix 1

DOI: https://doi.org/10.7554/eLife.43620.011

### Control for fluctuations in 1/f aperiodic signal

Periodic (i.e., oscillations) and aperiodic signals (i.e., 1/f 'background' noise: *Podvalny et al., 2015*; *Voytek et al., 2015*) constitute the total-band power spectrum estimated by FFT. Critically, it has been shown that changes in total-band power between different trial bins might not arise from a change in the periodic signal per se, but rather from a shift in the aperiodic signal via a change in either the signal offset, slope or both (*Haller et al., 2018*). On the one hand, an increase in the offset of the aperiodic signal may boost total-band power at all frequencies. On the other hand, an increase in aperiodic slope may manifest as a simultaneous increase in low-frequency total-band power and a decrease in high-frequency total-band power. A growing number of studies show that parameters of the aperiodic signal are related to cognitive and perceptual states (*Podvalny et al., 2015*), and are altered in aging (*Voytek et al., 2015*) and disease (*Peterson et al., 2017*). Importantly, it has been proposed that the aperiodic signal may reflect a physiological function (i.e., excitability/inhibition balance; *Voytek et al., 2015*) that is, at least partially, independent from the periodic signal.

For these reasons, we first set out to determine whether our original binning analysis based on total-band power (illustrated in *Figure 3*) separated trials based on periodic, aperiodic signal or both. To this end, we used the toolbox *fooof* (*Haller et al., 2018*; *Voytek et al., 2015*) to estimate the aperiodic-adjusted power, and the offset and slope of the aperiodic signal (at the frequency and channels of maximal statistical effects collapsed across conditions). We then compared (1) the aperiodic-adjusted power (*Figure 3D* and *Appendix 1—figure 1A*), (2) the aperiodic offset (*Appendix 1—figure 1B*); and (3) the slope (*Appendix 1—figure 1B*) of the power spectrum between trials of strongest ($Q_5$) and weakest ($Q_1$) total-band power (using paired-sample t-tests, corrected for multiple comparisons using FDR; *Benjamini and Hochberg, 1995*). We found that, compared to trials with weakest total-band power, trials with strongest total-band power were associated with an increase in aperiodic-adjusted power ($t(23) = 15.564$, FDR-corrected $p<0.001$, *Figure 3D* and *Appendix 1—figure 1A*). Specifically, the aperiodic-adjusted power on trials with strongest and weakest total-band power was 0.393 $\mu V_2$ 0.032 SEM, and 0.147 $\mu V_2$ 0.023 SEM. These results suggest that, in the binning analysis presented in *Figure 3*, trials of strongest total-band power were indeed associated with an increase in oscillations. In addition, we found that the power spectrum of trials with strongest total-band power was characterized by a higher offset ($t(23) = 10.091$, FDR-corrected $p<0.001$) and a steeper slope ($t(23) = 6.503$, FDR-corrected $p<0.001$, *Appendix 1—figure 1B*). Specifically, the offset on trials with strongest and weakest total-band power was 0.738 $\pm$ 0.017 SEM, and 0.619 $\pm$ 0.020 SEM, respectively; moreover, the slope on trials with strongest and weakest total-band power was 0.720 $\pm$ 0.016 SEM, 0.673 $\pm$ 0.020 SEM, respectively. These results implicate that trials of strongest and weakest total-band power not only differed in aperiodic-adjusted power, but also in the offset and slope of the aperiodic signal of the power spectrum. These results raise the question of whether the differences in ERP amplitude that we report in the analysis in *Figure 3A/B* may be due to differences in the aperiodic signal of the power spectrum (in addition to aperiodic-adjusted power).

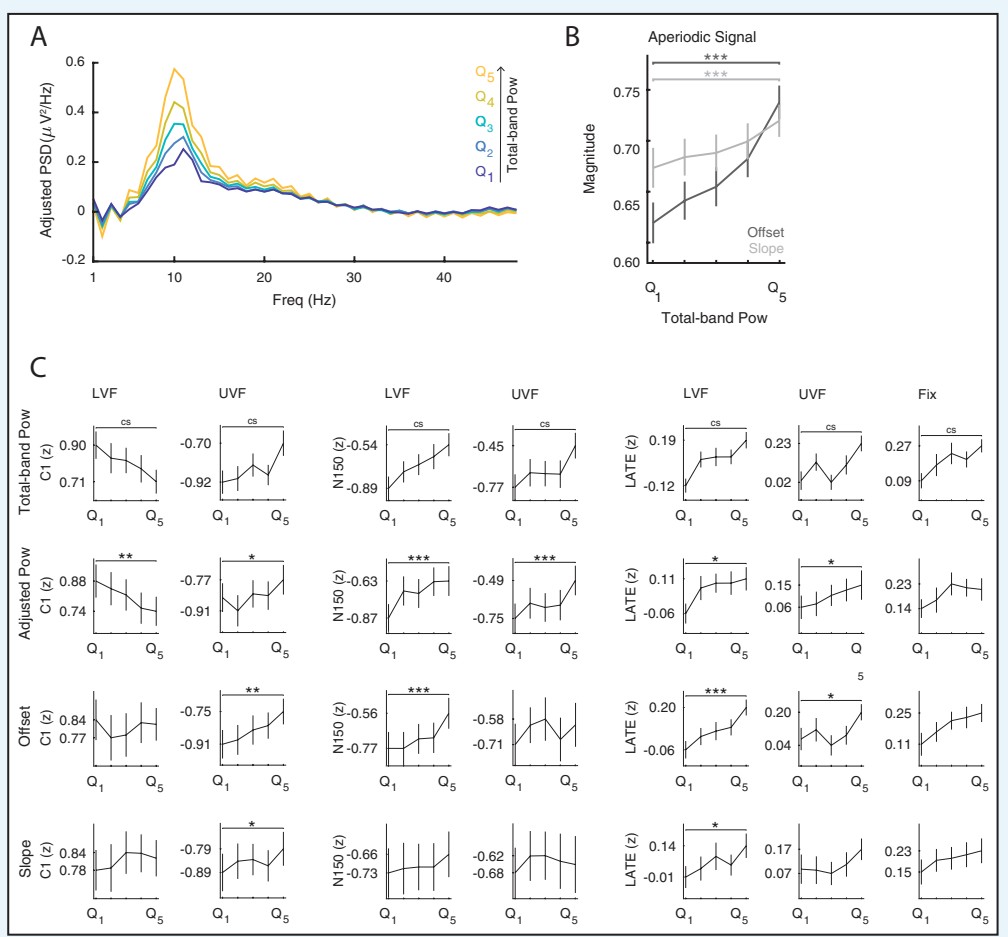

**Appendix 1—figure 1.** Analysis of periodic and aperiodic features of the prestimulus power spectrum. (**A**) Group-average prestimulus aperiodic-adjusted power spectrum shown separately for the five bins sorted from weak ($Q_1$) to strong ($Q_5$) total-band power. (**B**) Group-average offset (dark) and slope (light) of the aperiodic signal sorted from weak ($Q_1$) to strong ($Q_5$) total-band power. The results in A and B are shown for the frequencies and electrodes of most positive/negative t-statistics, collapsed across the significant clusters of **Figure 3**. (**C**) Group-average normalized ERP amplitude on trials sorted by total-band power, aperiodic-adjusted power, offset and slope. ERP amplitudes were compared across most extreme bins ($Q_{5-1}$) at the ERP time point and the prestimulus-power electrode and frequency of most positive/negative t-statistics, separately for each component and trial type. $***$, $**$, $*$ indicate FDR-corrected p-values<0.001, 0.01, 0.05, respectively. $cs$ indicates significant p-values based on cluster-level statistics (see **Figure 3**).

DOI: https://doi.org/10.7554/eLife.43620.012

To address this question, we examined whether the ERP amplitude differed between (1) bins of highest and lowest offset, and (2) bins of steepest and shallowest slope. To this end, we quantified a single-trial measure of the prestimulus power spectrum for the electrodes of maximal statistical effects separately for each component (C1, N150, and LATE), and for each trial type (LVF/UVF/Fix). Next, we classified trials in five bins based on single-trial estimates of the offset and the slope of the aperiodic signal (toolbox fooof: **Haller et al., 2018**), and quantified the ERP amplitude (at the subject-specific channels of maximal C1 response and at the time points of maximal statistical effects) for each bin. Then, we compared the ERP amplitude between bins of highest and lowest offset, and steepest and shallowest slope of the aperiodic signal (using paired-sample t-tests, corrected for multiple comparisons using FDR; **Benjamini and Hochberg, 1995**), **Appendix 1—figure 1C**)

A statistical test comparing the C1 on UVF trials of highest vs lowest offset revealed that the C1 component became weaker (i.e., less negative) on trials with highest offset (t(23) = 3.936 ; FDR-corrected p=0.003). A statistical test comparing the N150 on stimulation trials of highest vs lowest offset revealed that the N150 component became weaker (i.e., less negative) on trials with highest offset on LVF trials (t(23) = 4.640 ; FDR-corrected p<0.001), but not significantly on UVF trials (t(23) = 1.978; FDR-corrected p=0.077). In contrast, no significant relationship between ERP amplitude and offset was observed for the C1 on LVF trials (t(23) = −0.448 ; FDR-corrected p=0.658). Moreover, a statistical test comparing the late ERP amplitude on UVF trials of highest vs lowest offset revealed that the late ERP amplitude became stronger (i.e., more positive) on trials with highest offset in both LVF (t(23) = 4.851 ; FDR-corrected p<0.001) and UVF trials (t(23) = 2.683 ; p=0.037), but not significantly in Fix trials (t(23) = 2.193 ; FDR-corrected p=0.109).

A statistical test comparing the C1 on UVF trials of steepest vs shallowest slope revealed that the C1 component became weaker (i.e., less negative) on trials with steepest slope (t(23) = 2.492; FDR-corrected p=0.037). In contrast, no significant relationship between ERP amplitude and slope was observed for the C1 on LVF trials (t(23) = 1.150 ; FDR-corrected p=0.294), and for the N150 on both LVF (t(23) = 1.9024; FDR-corrected p=0.090) and UVF trials (t(23) = 0.628 ; p=0.536). Moreover, a statistical test comparing the late ERP amplitude of steepest vs shallowest slope on LVF trials revealed that the late ERP amplitude became stronger (i.e., more positive) on trials with steepest slope (t(23) = 2.432 ; FDR-corrected p=0.035). In contrast, no significant relationship between the late ERP amplitude and the slope was observed on UVF trials (t(23) = 1.422 ; FDR-corrected p=0.190), and Fix trials (t(23) = 1.560 ; FDR-corrected p=0.133).

The results of the aperiodic-based binning analysis demonstrate that the slope and offset are also related to ERP amplitude, although less consistently across components and trial types, compared to aperiodic-adjusted power. We believe that these mixed results might be due to the fact that the EEG signal is known to yield a noisy single-trial estimate of the power spectrum. While periodic features, such as $\alpha$-band oscillations, constitute the most prominent aspect of the EEG signal in humans, EEG estimates of the aperiodic features of the power spectrum may be more unreliable, especially on individual trials. To further corroborate the effects between ERP and aperiodic features, future studies should employ signals with higher signal-to-noise ratio (e.g., local-field-potentials) for a more reliable single-trial estimation of the aperiodic signal.

## Control for fluctuations in sleepiness

Previous studies on the relationship between prestimulus brain states and ERP amplitude have typically analyzed how differences in ERP amplitude were related to differences in prestimulus power across trials (e.g., *Başar and Stampfer, 1985*; *Jasiukaitis and Hakerem, 1988*; *Becker et al., 2008*). This across-trial approach treats individual trials as independent samples and therefore ignores the fact that data are collected in temporal order. This is potentially problematic, because it is known that both oscillatory power and ERP amplitude change over the course of an experiment, possibly due to a number of factors including progressive fatigue and sleepiness (*Boksem et al., 2005*). Specifically, prestimulus $\alpha$-band power is known to increase over the course of an experiment (*van Dijk et al., 2008*; *Mathewson et al., 2009*; *Benwell et al., 2017*; *Benwell et al., 2019*), suggesting increased inhibition, possibly as consequence of fatigue (*Kaida et al., 2006*). Likewise, the amplitude of early visual (N1: *Boksem et al., 2005*) and somatosensory (P60m: *Nikouline et al., 2000*) ERP components is known to change over the course of the experiment. In other words, since prestimulus power and ERP amplitude both covary across time (e.g., as a function of time-varying variables such as fatigue or sleepiness), their correlation could be epiphenomenal. Therefore, previous studies have tried to rule out that the across-trial correlation between ERP amplitude and power is confounded by fatigue by showing that power is not affected by time-varying temporal variables (e.g., trial number; *Nikouline et al., 2000*).

To address this issue, we collected subjective sleepiness ratings (see Stimuli and Experimental Design for details) at the end of each recording block. Then we used GLM to

estimate the correlation between these scores and prestimulus power for each electrode, frequency, and trial type . We used cluster permutation tests to determine whether this correlation (i.e., GLM $\beta$) was significantly different from 0 across participants. In all trial types, we found that subjective sleepiness was positively correlated with low-frequency power. Specifically, on LVF trials there were two significant positive clusters: the first cluster (p<0.001) spanned frequencies between 5 and 10 Hz in 63 electrodes with a centro-frontal peak (*Appendix 1—figure 2A*, top left panel); the second cluster (p=0.024) spanned frequencies between 13 and 18 Hz in 39 electrodes with a centro-frontal peak (*Appendix 1—figure 2A*, bottom left panel). On UVF trials there was one significant positive cluster (p=0.005) between 5 and 9 Hz in 64 electrodes with an occipital peak (*Appendix 1—figure 2A*, middle panel). On Fix trials there was one significant positive cluster (p<0.001) between 5 and 16 Hz in 62 electrodes with a central peak (*Appendix 1—figure 2A*, right panel). We then removed the contribution of sleepiness from the trial estimates of oscillatory power and repeated the binning analysis with these sleepiness-corrected measures of power ($P_{KSSC}$, see Influence of prestimulus oscillations on event-related potentials and oscillations for more details).

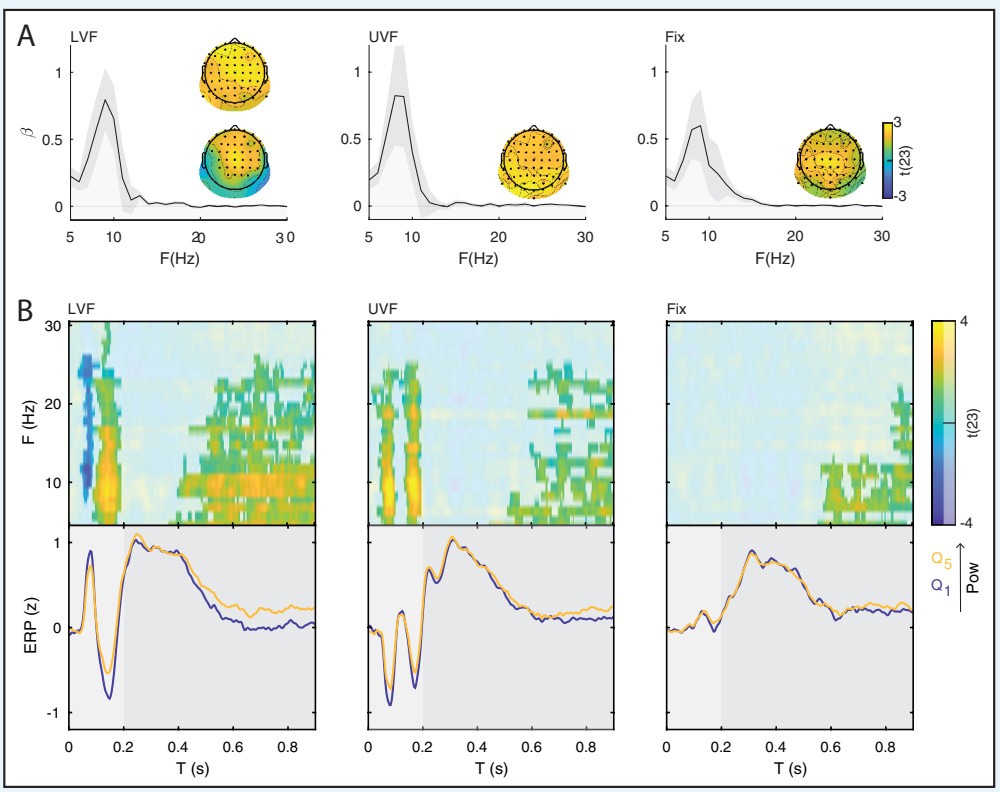

**Appendix 1—figure 2.** Interaction between ERP and prestimulus power when sleepiness effects are ruled out. (**A**) GLM was used to measure the contribution (regression coefficient; GLM $\beta$) of oscillatory power to subjective sleepiness ratings, obtained by the KSS questionnaire. The estimated GLM $\beta$s were tested against the null hypothesis of no relationship across the sample of participants using cluster permutation tests. The plots show the group-level GLM $\beta$ averaged across respective cluster electrodes, and per frequency. Shaded areas indicate $\pm$ SEM. The topographies show the positive t-statistics of the significant clusters averaged across the respective cluster frequencies. Positive values (yellow) indicate that power is positively correlated with KSS. (**B**) Group-level t-statistics maps of the difference in ERP amplitude between states of weak ($Q_1$) and strong ($Q_5$) prestimulus power for lower visual field stimuli (left panel, LVF), upper visual field stimuli (middle panel, UVF), and in fixation trials (right panel, Fix) . Positive (yellow) and negative t-statistics values (blue) indicate that ERP amplitude/voltage becomes more positive or more negative during states of strong prestimulus power, respectively. The maps are averaged across electrodes of the significant

cluster, and masked by a final alpha of 0.025 using separate two-sided cluster permutation testing for early and late ERP time windows. Note that the x-axis refers to post-stimulus ERP time, while the y-axis refers to the frequency of prestimulus oscillatory power. Bottom insets: visualization of the normalized ERP time course separately for states of strong ($Q_5$, yellow) and weak ($Q_1$, blue) prestimulus power. This was computed at the electrode and frequency of most positive/negative t-statistics during the C1 time window on stimulation trials and during the late time window on Fix trials. Time 0 s indicates stimulus/fixation-target onset. The results replicate the conventional analysis on raw power values reported in *Figure 3*.

DOI: https://doi.org/10.7554/eLife.43620.013

The effect of $P_{KSSC}$ on early and late ERP components (*Appendix 1—figure 2B*) was virtually identical to the previous analysis with raw power measures (*Figure 3A*). Specifically, on LVF trials, the statistical test during the early time window revealed a significant negative cluster (p=0.015) spanning frequencies between 8 and 26 Hz, time points between 0.043 and 0.121 s, and all 64 electrodes, and a significant positive cluster (p<0.001) spanning frequencies between 5 and 30 Hz, time points between 0.086 and 0.200, and all 64 electrodes. In addition, the statistical test during the late time window revealed one significant positive cluster (p<0.001) spanning frequencies between 5 and 26 Hz, time points between 0.371 and 0.900 s, and all 64 electrodes (*Appendix 1—figure 2B*, left panel). On UVF trials, the statistical test during the early time window revealed two significant positive clusters (p<0.001, *Appendix 1—figure 2B*, middle panel): the first cluster spanned frequencies between 5 and 25 Hz, time points between 0.020 and 0.117, and all 64 electrodes; the second cluster spanned frequencies between 5 and 25 Hz, time points between 0.121 and 0.200 s, and all 64 electrodes. In addition, the statistical test during the late time window revealed one significant positive cluster (p<0.008) spanning frequencies between 5 and 17 Hz, time points between 0.449 and 0.902 s, and 62 electrodes. There was also a near-significant positive cluster (p=0.035) at frequencies between 17 and 25 Hz, time points between 0.590 and 0.900 s, and at 62 electrodes. On Fix trials, no significant cluster was found during the early time window. The statistical test during the late time window revealed one significant positive cluster (p=0.003) spanning frequencies between 5 and 22 Hz, time points between 0.559 and 0.900 s, and all 64 electrodes (*Appendix 1—figure 2B*, right panel).

Taken together, these results indicate that the relationship between prestimulus power and ERP amplitude is unlikely to be determined by time-varying effects driven by spontaneous fluctuations of fatigue or sleepiness.

