## [Decision Letter]

Thank you for submitting your article "Multiple mechanisms link prestimulus neural oscillations to sensory responses" for consideration by *eLife*. Your article has been reviewed by two peer reviewers, one of whom is a member of our Board of Reviewing Editors, and the evaluation has been overseen by and Laura Colgin as the Senior Editor. Both reviewers have opted to remain anonymous.

The reviewers have discussed the reviews with one another and the Reviewing Editor has drafted this decision to help you prepare a revised submission.

Summary:

Both reviewers find that your study describes interesting new insights as to how

pre-stimulus alpha/beta power affects the magnitude of early and late visually evoked neural responses. The main finding is that levels of oscillatory power in the prestimulus period have different effects on early and late stimulus-evoked responses. On the one hand, the results show a positive correlation between prestimulus power and late ERP components, caused by asymmetric (i.e. non-zero) shifts of oscillations. On the other hand, the results show a negative relation between prestimulus alpha/beta power and early ERP components, which is consistent with the notion of an inhibitory role of alpha.

Essential revisions:

Both reviewers, while being generally supportive of the manuscript, raise a number of concerns that must be adequately addressed before the paper can be accepted. Some of the required revisions will require further analysis, while others require modifications of the figures and/or text.

1) Both reviewers highlighted that trials were sorted according to the amount of prestimulus power in a very broad range (8-30Hz). This raises the question as to whether the effects are driven by alpha oscillations as opposed to changes in the 1/f spectrum. In order to address this point the authors should redo their analysis by controlling for fluctuations in the 1/f spectrum, for instance by subtracting 1/f on a trial-by-trial level (see Voytek et al., 2015; J Neurosci; or Podvalny et al., 2015; J Neurophysiol for possible methods).

2) One of the reviewers highlighted a complete lack of discussion of relevant literature. This refers to a substantial body of evidence demonstrating that early evoked ERP components are generated by a phase-reset of ongoing oscillations (i.e Sayers et al. 1974; Nature; Makeig et al., 2002; Science; Gruber et al., 2005; Cerebral Cortex; Hanslmayr et al., 2007; Cerebral Cortex; Sauseng et al. 2007; Neurosci Biobheav Rev). The authors should include a thorough discussion of this literature in relation to their findings.

3) Circularity in analysis. The correlation between prestimulus power and ERD appears to be highly circular since the prestimulus power is used also to estimate the poststimulus response. In order to address this point the authors should either drop this analysis entirely, or redo the analysis in a way that completely avoids circularity.

4) One reviewer noted that the use of '<' and '>' symbols as detection targets seemed odd and might have produced involuntary shifts of attention. Please include a discussion of whether these effects could be a possible confounding factor.

[Editors' note: further revisions were requested prior to acceptance, as described below.]

Thank you for resubmitting your work entitled "Multiple mechanisms link prestimulus neural oscillations to sensory responses" for further consideration at *eLife*. Your revised article has been favorably evaluated by Laura Colgin (Senior Editor), and two reviewers, one of whom is a member of our Board of Reviewing Editors.

The manuscript has been improved but there is one remaining point, raised by reviewer #1 referring to the circular analysis of alpha power. This point should be clearly addressed in the Results section.

Reviewer #1:

The authors have done a thorough job in addressing all of my concerns. A central concern has been the 1/f issue which the authors took to heart and provided convincing evidence against this concern. I do not 100% percent agree with their statements on phase-reset, in particular the argument that a positive relation between alpha power and the late ERP rules out phase reset. Since Alpha is indeed a much faster oscillation than the one reflected in the late ERP (i.e. delta) the phase-reset theory would not make a prediction about any relationship between alpha and this late component. The fact that they do find a correlation supports the idea of the baseline-shift, but that doesn't mean that the early components can't be generated by phase-reset. However, the authors may agree or not with this statement and that is fine.

Reviewer #2:

The authors have responded satisfactorily to all my concerns.

---

## [Author Response]

Essential revisions:Both reviewers, while being generally supportive of the manuscript, raise a number of concerns that must be adequately addressed before the paper can be accepted. Some of the required revisions will require further analysis, while others require modifications of the figures and/or text.1) Both reviewers highlighted that trials were sorted according to the amount of prestimulus power in a very broad range (8-30Hz). This raises the question as to whether the effects are driven by alpha oscillations as opposed to changes in the 1/f spectrum. In order to address this point the authors should redo their analysis by controlling for fluctuations in the 1/f spectrum, for instance by subtracting 1/f on a trial-by-trial level (see Voytek et al., 2015; J Neurosci; or Podvalny et al., 2015; J Neurophysiol for possible methods).

We thank the reviewers for pointing out this very relevant issue. We have now added several analyses to address this point, and believe it has further substantiated and improved our original findings. To recapitulate, the main finding of our study was that the trial-by-trial variability of the ERP amplitude can be partly explained by the power of prestimulus oscillations in a broad frequency band, including the *α-* and *β-*band rhythms. Specifically, we estimated power using the fast Fourier transform as a mixture of a periodic signal (i.e., oscillations) and an aperiodic signal, identified by its characteristic power-law 1/*f^x^* shape (i.e., power-law decrease in the power spectrum as a function of frequency *f*, where the exponent *x* determines the slope of the decline). We originally did not dissociate between these two contributions, and used this total-band estimate of power to classify trials into strong and weak power bins.

Critically, it has been shown that changes in total-band power between different trial bins might not arise from a change in the periodic signal per se, but rather from a shift in the aperiodic signal via a change in either the signal offset, its slope, or both. On the one hand, an increase in the offset of the aperiodic signal may boost total-band power at all frequencies. On the other hand, an increase in aperiodic slope may manifest as a simultaneous increase in low-frequency total-band power and a decrease in high-frequency total-band power. Therefore, it is possible that our analysis did not specifically separate trials based on actual oscillatory power, but rather on the features of the underlying aperiodic signal. Accordingly, we agree with the reviewers that differences in ERP amplitude between bins of strong and weak total-band power (illustrated in Figure 3) might be explained by a change in the aperiodic signal (slope or offset), rather than a genuine increase in oscillatory power.

We first set out to determine whether our original binning analysis based on total-band power (Figure 3) separated trials based on periodic signal, the aperiodic signal, or both signals. To this end, we used the toolbox *fooof* released by the Voytek lab (Haller et al., 2018) to estimate the aperiodic signal from the original total-band power spectrum, separately for each bin of total-band power. Specifically, this algorithm fits the total-band power spectrum with an aperiodic function defined by a slope and an offset (i.e., aperiodic signal). To obtain a measure of the periodic signal, we subtracted this aperiodic function from the original power spectrum, resulting in an aperiodic-adjusted or flattened power spectrum. We estimated the aperiodic-adjusted power at the frequency and channels of maximal effects (collapsed across all significant clusters). We then compared (1) the total-band power, (2) the aperiodic-adjusted power, (3) the aperiodic offset, and (4) the slope of the power spectrum between trials of strongest and weakest total-band power. We found that, compared to trials of weakest total-band power, trials of strongest total-band power were associated with an increase in aperiodic-adjusted power. This result suggests that, in the binning analysis presented in Figure 3 in the original manuscript, trials of strongest total-band power were indeed associated with a genuine increase in oscillatory power. In addition, we found that the power spectrum of trials of strongest total-band power was characterized by a higher offset and a steeper slope. These results suggest that trials of strongest and weakest total-band power not only differed in aperiodic-adjusted power (i.e., actual oscillatory power), but also in the offset and slope of the aperiodic signal of the power spectrum. Consistent with the reviewers’ concerns, these results raise the question of whether the differences in ERP amplitude that we report are due to differences in actual oscillatory power or rather in the aperiodic signal.

To address this concern, we followed the reviewers’ suggestion and replicated the results of the binning analysis using aperiodic-adjusted power estimates, rather than the total-band power. To this end, we quantified a single-trial measure of total-band prestimulus power for the frequency and electrode of maximal effects separately for each component (C1, N150, and LATE), and for each trial type (LVF/UVF/Fix), as reported in the original manuscript. Then, we estimated a single-trial measure of the offset and slope of the aperiodic signal, and of the aperiodic-adjusted power using the toolbox fooof (Haller et al., 2018). Following the same procedure described in the original manuscript, we classified the trials in five bins based on single-trial estimates of (1) the aperiodic-adjusted power, (2) the offset and (3) the slope of the aperiodic signal, and quantified the ERP amplitude (at the subject-specific channels of maximal C1 response) for each bin. Then, we compared the ERP amplitude at the time points of maximal effect separately for each component, and for each trial type between bins of (1) strongest and weakest aperiodic-adjusted power, (2) highest and lowest offset, and (3) steepest and most shallow slope of the aperiodic signal. We summarized the results in panel B of Appendix 1—figure 1.

A statistical test comparing the C1 amplitude of strongest vs weakest aperiodic-adjusted power revealed that the C1 became weaker (i.e., less positive on LVF trials or less negative on UVF trials) on trials of strong prestimulus power. Likewise, a statistical test comparing the N150 amplitude of strongest vs weakest aperiodic-adjusted power revealed that the N150 became significantly weaker (i.e., less negative on stimulation trials) on trials of strong prestimulus power. Taken together, the results for the early ERP time window replicate the results shown in Figure 3 of the original manuscript.

In addition, a statistical test comparing the late ERP amplitude of strongest vs weakest aperiodic-adjusted power revealed that the late component became stronger (i.e., more positive in all trials) on trials of strong prestimulus power for the stimulation condition, and less consistently for the fixation-only condition. Taken together, the results on the late ERP time window largely replicate the results shown in Figure 3 of the original manuscript.

In summary, the effects on both early and late ERP components based on total-band power reported in Figure 3, are replicated by an analysis that controls for fluctuations in the 1/f aperiodic signal, demonstrating that the ERP amplitude was indeed modulated by actual oscillatory power.

We have described the results of the analysis of aperiodic-adjusted power in the Results of the revised manuscript as follows:

“The functional inhibition account predicts that states of strong prestimulus power reflect neural inhibition, resulting in reduced amplitude specifically of early ERP components generated by the additive mechanism. […] The most positive t-statistic was found at 10 Hz, 0.168 ms, and at electrode F2 (total-band power: t(23) = 8.544; aperiodic-adjusted power: t(23) = 5.785; FDR-corrected p < 0.001, Appendix 1—figure 1C).”

We have included a discussion of these results in the Discussion section of the revised manuscript:

“Additionally, previous results may have also have been due to (1) fluctuations of the 1/*f* aperiodic signal (which affect total-band power estimates, see Appendix 1—figure 1), or (2) fluctuations of sleepiness (which affect both oscillatory power and ERP amplitude, see Appendix 1—figure 2). […] In the current study, we confirmed that the late ERP amplitude was indeed amplified by oscillatory power, rather than just the 1/*f* aperiodic signal, and that this effect was not an epiphenomenon due to sleepiness.”

We have included a description of the analysis of aperiodic-adjusted power in the Materials and methods of the revised manuscript as follows:

“Because the estimates of prestimulus power were computed with an FFT, they reflect a mixture of periodic (i.e., oscillations) and aperiodic signals (i.e, 1/f “background” noise: Podvalny et al., 2015; Voytek et al., 2015), referred to as total-band power. […] Then, we compared the ERP amplitude between bins of strongest and weakest aperiodic-adjusted power (using paired-sample t-tests, corrected for multiple comparisons using FDR; Benjamini and Hochberg, 1995).”

In addition to binning based on the aperiodic-adjusted power, we further explored the data by binning the trials based on single-trial estimates of the aperiodic features (i.e., offset and slope) of the power spectrum. We summarized the results in Figure SI1B. The results of the aperiodic-based binning analysis demonstrate that the slope and offset are also related to ERP amplitude, although less consistently across components and trial types compared to oscillatory power. We believe that these mixed results might be due to the fact that the EEG signal is known to yield a noisy single-trial estimate of the power spectrum. While periodic features, such as *α*-band oscillations, constitute the most prominent aspect of the EEG signal in humans, EEG estimates of the aperiodic features of the power spectrum may be more unreliable, especially in individual trials. To further corroborate the effects between ERP and aperiodic features, future studies should employ signals with higher signal-to-noise ratio (e.g., local-field-potential) for a more reliable single-trial estimation of the aperiodic signal.

In addition, we have described the methods and results of the analysis of the aperiodic features of the power spectrum in Appendix 1 (section Control for fluctuations in 1/*f* aperiodic signal) of the revised manuscript as follows:

“Periodic (i.e., oscillations) and aperiodic signals (i.e., 1/f“background” noise: Podvalny et al., 2015; Voytek et al., 2015) constitute the total-band power spectrum estimated by FFT. […] To further corroborate the effects between ERP and aperiodic features, future studies should employ signals with higher signal-to-noise ratio (e.g., local-field-potentials) for a more reliable single-trial estimation of the aperiodic signal.”

2) One of the reviewers highlighted a complete lack of discussion of relevant literature. This refers to a substantial body of evidence demonstrating that early evoked ERP components are generated by a phase-reset of ongoing oscillations (i.e Sayers et al. 1974; Makeig et al., 2002; Gruber et al., 2005; Hanslmayr et al., 2007; Sauseng et al. 2007). The authors should include a thorough discussion of this literature in relation to their findings.

We apologize for this oversight, and thank the reviewer for pointing us to this relevant literature. In the revised version of the manuscript, we provide a review of the literature on phase reset and discuss how it relates to our current findings. Although our experiment was not designed to specifically test the phase-reset hypothesis, we see two possible predictions that a phase-reset account could make regarding the relationship between prestimulus oscillatory power and ERP amplitude (Hanslmayr et al., 2007; Sauseng et al., 2007).

On the one hand, it has been argued that phase reset in response to a stimulus can only occur if the oscillation already exists prior to the reset; i.e., during the prestimulus window. It follows that any ERP component generated by phase reset is likely to be absent during desynchronized states (i.e., weakest power trials; Shah et al., 2004; Sauseng et al., 2007). Moreover, the phase reset of an oscillation of a particular frequency is expected to generate a component with similar frequency characteristic: e.g., the *α*-band phase reset is thought to generate early ERP components of the same polarity with an inter-peak latency at approximately 100 ms (as the C1 and N150 on UVF trials, see Figure 2B in the main manuscript; Hanslmayr et al., 2007; Sauseng et al., 2007). These considerations suggest that trials of weakest prestimulus power may result in less prominent phase reset, which would manifest as a reduction of ERP amplitude. Accordingly, we would expect a positive relationship between prestimulus oscillations in the *α*- and *β*-bands and the C1 and N150 amplitude on UVF trials, which occur with an inter-peak latency of approximately 70–80 ms. Contrary to this prediction, our study showed that the amplitude of the C1 and N150 components on UVF trials was negatively correlated with prestimulus *α*- and *β*-band oscillations. In contrast, the amplitude of the late ERP component was positively correlated with *α*- and *β*-band oscillations, consistent with the predicted directionality of the phase-reset effect. However, this effect is unlikely to be due to phase reset because of its frequency-specificity: in fact, according to a phase-reset account, a sustained late component with a latency longer than 200 ms is expected to be generated only by an ongoing oscillation below 5 Hz. Therefore, the effect of prestimulus *α*- and *β*-band oscillations on the late ERP components is better explained by baseline shift, rather than phase reset.

On the other hand, one can argue that strong oscillations represent a state with pronounced neuronal synchronization that is not easily affected by the weak inputs, as also shown in a previous modeling study (Hansel and Sompolinsky, 1996). Therefore, during states of strong ongoing oscillations, phase- reset may be harder to be achieved and, thus, is unlikely to result in the generation of an ERP component. Accordingly, this predicts a negative relationship between prestimulus power and ERP amplitude, consistent with our results and with the prediction of the functional inhibition account. In contrast, the positive relationship between prestimulus power and the late ERP component does not support this prediction from the phase-reset mechanism. In summary, we believe that the findings reported in our study are better accounted for by a modulation of the additive mechanism (via functional inhibition) as well as by the baseline-shift mechanism, rather than phase reset.

We have included an in-depth review of the literature on phase reset in relation to our findings, in the Discussion of the revised manuscript (section What about phase reset?) as follows:

“In this study we considered the additive (Bijma et al., 2003; Shah et al., 2004; Mäkinen et al., 2005; Mazaheri and Jensen, 2006) and baseline-shift (Nikulin et al., 2007; Mazaheri and Jensen, 2008) mechanisms for the generation of early and late ERP components (Bijma et al., 2003; Shah et al., 2004; Mäkinen et al., 2005; Mazaheri and Jensen, 2006), respectively. […] In contrast, the positive relationship between prestimulus power and the late ERP component does not support this prediction from the phase-reset account. In summary, the results of this study are better accounted for by a modulation of the additive mechanism (via functional inhibition) as well as by the baseline-shift mechanism, rather than phase reset.”

3) Circularity in analysis. The correlation between prestimulus power and ERD appears to be highly circular since the prestimulus power is used also to estimate the poststimulus response. In order to address this point the authors should either drop this analysis entirely, or redo the analysis in a way that completely avoids circularity.

We appreciate the reviewers’ comments and we would like to discuss the issue in detail below. We conducted this analysis to demonstrate that prestimulus power is statistically correlated with poststimulus ERD, which in turn is statistically correlated with the amplitude of the late ERP. We agree that the strong correlation between prestimulus power and ERD is to some extent trivial. However, conducting this analysis is necessary for testing a prediction of the baseline-shift hypothesis, i.e., that the relationship between prestimulus power and the late ERP is due to the ERD of non-zero-mean oscillations and to conventional signal processing procedures (i.e., baseline correction), rather than due to a modulation of an additive response. To recap, in this analysis, we compared the ERD magnitude between groups of trials of weakest and strongest prestimulus power. Importantly, we computed the ERD at a specific frequency by subtracting the average prestimulus power from the time course of post-stimulus power. The results of this analysis show that states of strong prestimulus power are associated with strong ERD magnitude, consistent with the results of several previous studies that we discussed in the original manuscript (Min et al., 2007; Becker et al., 2008; Tenke et al., 2015; Benwell et al., 2017).

We agree with the reviewers that the strong correlation between prestimulus power and ERD is statistically circular because the dependent variable (i.e., prestimulus power) was used to compute the independent variable (i.e., ERD). To address this issue, we have removed the testing for statistical significance of the relationship between prestimulus power and ERD magnitude in the revised manuscript. Although the statistical estimates of this relationship would be inflated (due to circularity), we believe the pattern is nevertheless important in that it is directly predicted by the baseline-shift mechanism. Specifically, we found that *α*-band power is reduced to approximately the same level regardless of prestimulus power (Figure 6C). Accordingly, whereas the average prestimulus voltage is expected to differ between different prestimulus power bins due to the non-zero-mean property of neural oscillations, the average post-stimulus voltage in the late window is expected to be the same regardless of prestimulus power (Figure 1, right panel). The baseline-shift mechanism predicts that subtracting a stronger prestimulus signal (strong power bin) yields a stronger shift of the EEG signal from the prestimulus baseline, and thus a stronger late ERP component. In summary, we believe that the results of the analysis illustrated in Figure 6 of the manuscript will help the reader better understand the baseline-shift mechanism and we would like to include a visualization of this relationship in the revised manuscript.

In the revised manuscript, we have re-phrased the description of the results illustrated in Figure 6 in the Results as follows:

“After demonstrating that the ERD magnitude correlates with the late ERP amplitude, we determined whether the ERD magnitude was, in turn, related to prestimulus power, as predicted by the baseline-shift account. […]We found that strong low-frequency prestimulus oscillations were associated with strong ERD in all trial types (Figure 6).”

To address the reviewers’ concern about circularity, we added a section in the Discussion as follows:

“The baseline-shift account predicts that stronger ERD occurs during states of stronger prestimulus power, which generates a greater baseline shift. In the case of negative non-zero-mean oscillations, this process results in an enhancement of the late ERP component with positive polarity. […] The baseline-shift account predicts that subtracting a stronger prestimulus signal (strong power bin) yields a stronger shift of the EEG signal from the prestimulus baseline, and thus a stronger late ERP component (Figure 1B, lower panel).”

4) One reviewer noted that the use of '<' and '>' symbols as detection targets seemed odd and might have produced involuntary shifts of attention. Please include a discussion of whether these effects could be a possible confounding factor.

To address the reviewer’s concern, in the revised manuscript we have included a discussion of the detection targets used in the study and of possible confounding effects that they may have introduced. To recapitulate, in our study we added a discrimination task at the central fixation mark to ensure that the participants maintained the gaze to the center during stimulus presentation. Specifically, we showed either one of two equally probable central targets (“>” or “<”) during stimulus presentation (or at an equivalent time in fixation-only trials) and asked the participants to identify the target. It is possible that: first, because these targets pointed to either one of the two lateral visual hemifields, they might have caused involuntary shifts of spatial attention; and second, that these shifts might have confounded the effects of prestimulus oscillations, which are known to be modulated by spatial attention. It is important to note that it was impossible for participants to prepare and selectively attend either one of the lateral hemifields during the prestimulus window because the two targets were equally probable in each trial (each recording block included 50% “<” targets and 50% “>” targets). So, even if the attentional shifts occurred in the post-stimulus window, they were unlikely to confound the prestimulus effects reported in this study because they would cancel out across trials of different levels of prestimulus power. Furthermore, given that the visual stimuli used in this study were presented either in the lower or upper visual field, lateral shifts in spatial attention were unlikely to affect the vertical distribution of attention. Again, even if the attentional shifts modulated ERPs, the fully randomized presentation within each block ensured that these effects would cancel out across trials.

In summary, we believe that, due to the experimental design, the targets were unlikely to confound the prestimulus effects reported in this study. We agree with the reviewer that this is an important issue to discuss and therefore we have revised the Materials and methods of the manuscript as follows:

“On stimulation trials the central fixation mark turned into either one of two equally probable targets (“>” or “<”) during stimulus presentation for a duration of 0.100 s. On Fix trials, the change of the central fixation mark into the target occurred during a 0.100 s window between 1.8 and 2.4 s relative to trial onset. Note that, while the targets might have caused an involuntary shift of lateral attention, these effects would have cancelled out across trials because of the fully randomized presentation of the targets (each recording block included 50%“<” targets and 50%“>” targets).”

[Editors' note: further revisions were requested prior to acceptance, as described below.]

The manuscript has been improved but there is one remaining point, raised by reviewer #1 referring to the circular analysis of alpha power. This point should be clearly addressed in the Results section.Reviewer #1:The authors have done a thorough job in addressing all of my concerns. A central concern has been the 1/f issue which the authors took to heart and provided convincing evidence against this concern. I do not 100% percent agree with their statements on phase-reset, in particular the argument that a positive relation between alpha power and the late ERP rules out phase reset. Since Alpha is indeed a much faster oscillation than the one reflected in the late ERP (i.e. delta) the phase-reset theory would not make a prediction about any relationship between alpha and this late component. The fact that they do find a correlation supports the idea of the baseline-shift, but that doesn't mean that the early components can't be generated by phase-reset. However, the authors may agree or not with this statement and that is fine.

We thank the reviewer for the useful feedback on the discussion of our results. The reviewer correctly points out that the phase reset of an oscillation at a particular frequency is expected to generate a component with similar frequency characteristics: accordingly, it is not possible that phase reset of *α*-band oscillations is reflected in the slow ERP component. To address the reviewer’s comment, we have modified the section entitled “What about phase reset?” in the Discussion of the revised manuscript as follows:

“In addition, we found that the amplitude of the slow ERP component was positively correlated with prestimulus *α-* and *β*-band oscillations. However, this correlation is unlikely to be accounted for by phase reset of *α*- and *β*-band oscillations because these rhythms are much faster than the ones reflected in the slow ERP component (i.e., δ rhythm).”

In addition, the reviewer highlights that our results cannot fully distinguish between additive and phase-reset mechanisms of ERP generation during the early time window. We agree with this point and have now commented on this issue at the end of the section entitled “What about phase reset?” in the Discussion of the revised manuscript as follows:

“On the other hand, one can argue that strong oscillations represent a state with pronounced neuronal synchronization that is not easily affected by weak sensory inputs, as also shown in previous modeling work (Hansel and Sompolinsky, 1996). Thus, during states of strong ongoing oscillations, phase-reset may be harder to be achieved and, consequently, is unlikely to result in ERP generation. Accordingly, this predicts an ERP attenuation by prestimulus power, consistent with our results during the early time window. It is worth noting that, in this study, it is difficult to distinguish whether this attenuation affects ERP components generated by additive or phase-reset mechanisms; invasive electrophysiological recordings allowing for higher spatial resolution might be required to address this particular question (Hanslmayr et al., 2007; Telenczuk et al., 2010). Regardless of the underlying mechanisms of ERP generation, our results during the early time window can be explained by the functional inhibition account.”